# Piezo robotic hand for motion manipulation from micro to macro

Shijing Zhang [1,2], Yingxiang Liu [1,2] ✉, Jie Deng[1,2], Xiang Gao[1], Jing Li [1], Weiyi Wang[1], Mingxin Xun[1], Xuefeng Ma[1], Qingbing Chang[1], Junkao Liu[1], Weishan Chen[1] & Jie Zhao[1]

Multiple degrees of freedom (DOFs) motion manipulation of various objects is a crucial skill for robotic systems, which relies on various robotic hands. However, traditional robotic hands suffer from problems of low manipulation accuracy, poor electromagnetic compatibility and complex system due to limitations in structures, principles and transmissions. Here we present a direct-drive rigid piezo robotic hand (PRH) constructed on functional piezoelectric ceramic. Our PRH holds four piezo fingers and twelve motion DOFs. It achieves high adaptability motion manipulation of ten objects employing pre-planned functionalized hand gestures, manipulating plates to achieve 2L (linear) and 1R (rotary) motions, cylindrical objects to generate 1L and 1R motions and spherical objects to produce 3R motions. It holds promising prospects in constructing multi-DOF ultra-precision manipulation devices, and an integrated system of our PRH is developed to implement several applications. This work provides a new direction to develop robotic hand for multi-DOF motion manipulation from micro scale to macro scale.

The increasing number of robotic systems has promoted rapid progress in some advanced technology fields, including biomedical engineering[1,2], clinical surgery[3–5], semiconductor manufacturing[6,7], micro/nano positioning and manipulating[8,9]. Meanwhile, multi-DOF motion manipulation of various objects has been a crucial skill for robotic systems[10,11]. The robotic hand, which can replace the human hand to achieve dexterous manipulations or accomplish human–machine collaboration tasks, has been a significant research topic[12,13]. It has been usually utilized as the execution end of the robotic system. The functions and performance of the robotic hand determine the service capability of the entire robotic system. In short, the development of the robotic hand has been essential for the progress of robot-assisted manipulating technology[14,15].

In recent decades, various successful designs of robotic hands have been reported. The existing robotic hands can be classified into several representative kinds according to their differences in structural features, driving principles, and transmission mechanisms (Fig. 1a). Above all, according to structural features, the existing robotic hands

can be roughly classified into soft robotic hand[16–19], rigid robotic hand[20], and hybrid robotic hand[21]. Among them, the soft robotic hand means that its fingers and main structure are made of flexible materials, rather than rigid materials[22,23]; it holds obvious advantages of high flexibility, high adaptability, and strong cushioning characteristics[24–27]. Nevertheless, the challenges in the design, manufacturing, and control of flexible structures are inherited by the soft robotic hand[28]. The rigid robotic hand is made of rigid components and the whole system has relatively high rigidity[29]. It not only has the ability to resist large impact deformation but also is easy to realize relatively high manipulation accuracy and large manipulation force. Some researchers also design the robotic hand with a rigid and soft hybrid scheme[21]. It is named a hybrid robotic hand, in which the components of its core skeleton are designed to be rigid, while the exterior is wrapped with soft materials[30]. Next, the existing robotic hands can be mainly divided into electromagnetic, pneumatic, and functional material robotic hands according to the driving principles. The electromagnetic one means that the electromagnetic motor is used to convert electric energy into

[1]State Key Laboratory of Robotics and System, Harbin Institute of Technology, 150001 Harbin, China. [2]These authors contributed equally: Shijing Zhang, Yingxiang Liu, Jie Deng. ✉e-mail: liuyingxiang868@hit.edu.cn

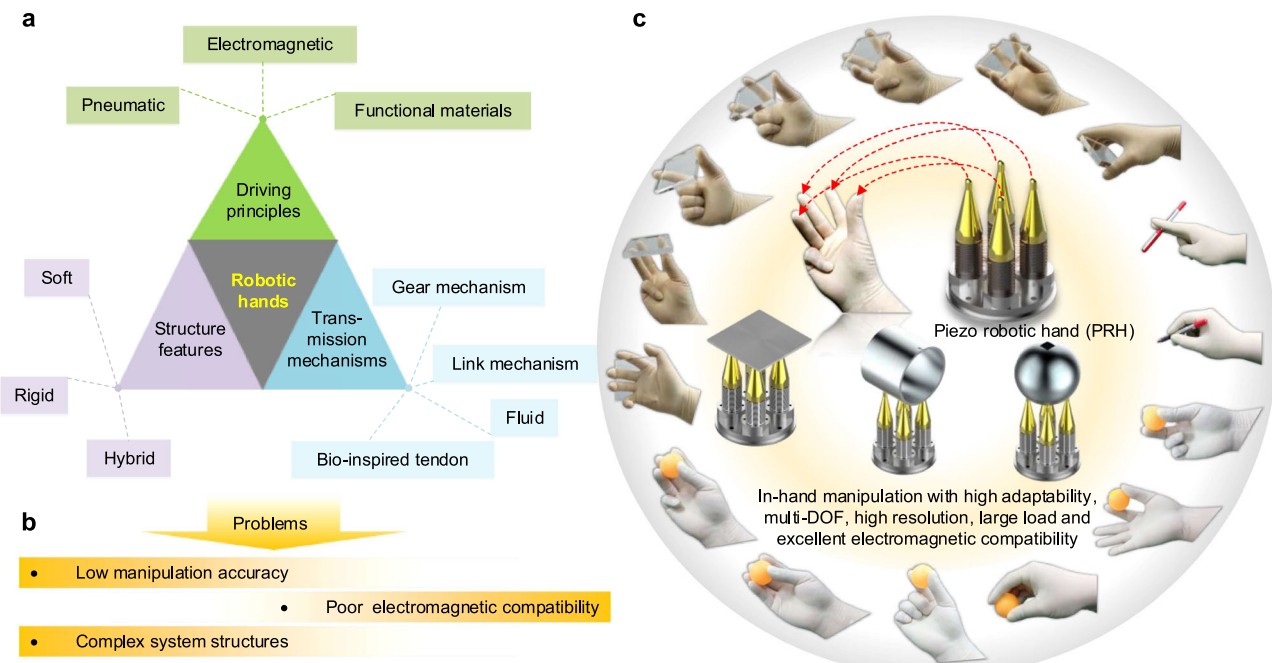

**Fig. 1 | Classifications and problems of the existing robotic hands, as well as the piezo robotic hand (PRH) inspired by in-hand motion manipulation.** **a** Classifications of typical robotic hands according to their structure features, driving principles, and transmission mechanisms. **b** Several typical problems of the existing robotic hands. **c** Our work to solve the problems of current robotic hands: a unique PRH inspired by in-hand manipulation of the human hand, which is available for motion manipulation with advantages of high adaptability, multi-DOF, high resolution, large load, and excellent electromagnetic compatibility.

mechanical energy[31]. It has the advantage of being easy to control, but poor electromagnetic compatibility is a typical defect. The pneumatic robotic hand usually uses gas as the medium to stimulate the pneumatic actuator and perform energy conversion from fluid kinetic energy to mechanical energy. The air cylinder and air chamber integrated into the robotic hand are inflated and deflated alternately; thus, the robotic hand achieves motion manipulation by deformations of the finger or the main body[32,33]. This type of robotic hand has high requirements for air tightness, and it is a challenge to achieve accurate motion control due to the compressibility of the gas medium; the external air source, pneumatic pumps, and control valves are usually necessary[34,35]. Besides, some functional materials are also used to develop robotic hands, for example, multifunctional artificial muscles[36,37]; their manipulation accuracy and response velocity are limited by slow electro-thermal conversion and cooling process. In addition, the existing robotic hands utilize several typical transmission mechanisms to transform motion, force, or torque. They mainly contain gear mechanism[38], link mechanism[39], bio-inspired tendon[40], and fluid medium transmission schemes[41]. Specifically, the gear and link mechanisms are usually combined with electromagnetic motors. The bio-inspired tendon transmission scheme usually uses the thread, rope, and pulley components to simulate the stretching and contracting motions of the bio-tendon[42]. Besides, the fluid transmission scheme means that the robotic hand utilizes fluid as the medium to realize motion transmission.

To sum up, the existing robotic hands have several representative problems caused by their structures, principles, and transmissions (Fig. 1b): (i) the manipulation accuracies of the current robotic hands are generally poor due to the low accuracy of the drive components and the transmission error; (ii) the integration of electromagnetic devices in the robotic hand leads to poor electromagnetic compatibility; and (iii) the structures of the current robotic hand are usually complex due to the use of transmission mechanisms. These problems always limit the applications of current robotic hands in motion manipulation. We have noted that these problems are mainly caused by energy conversion principles

and transmission mechanisms. Specifically, the use of drive components based on electromagnetic effect leads to the inevitability of electromagnetic interference; the utilization of complex transmission mechanisms increases the system complexity of the robotic hand; the low accuracies of the drive components lead to the low manipulation accuracy of the robotic hand. In summary, how to find new drive modes with other energy conversion principles and excellent electromagnetic compatibility is a challenging task. Meanwhile, how to design more efficient and accurate transmission mechanisms or completely abandon the transmission mechanism is a significant design idea for developing robotic hands. Besides, the robotic hand needs to achieve fast response, large output force, multi-DOF, and high adaptability for the requirements of advanced technology applications.

Aiming at the above problems, we innovatively propose a unique direct-drive rigid PRH inspired by in-hand motion manipulation (Fig. 1c). The main novelties and contributions of this work can be concluded as: (i) This work proposes the first robotic hand with four fingers constructed on piezoelectric ceramics, it achieves in-hand multi-DOF manipulation, as well as high resolution of 15 nm, fast response of 0.5 ms, low hysteresis of 3.95% and no electromagnetic interference. (ii) The PRH is designed to be a unique rigid configuration with four bolt-clamped metal-ceramic sandwich piezo fingers, which achieves compact structures and large load ability as it has no transmission mechanisms and hinges. (iii) The PRH achieves motion expansion from the micro deformations of piezoelectric ceramics to the macro motions of various objects by four-finger cooperative manipulations and realizes multi-DOF and cross-scale motion manipulation. (iv) The PRH realizes excellent adaptability to manipulate various objects with diverse shapes, materials, and dimensions. A series of experiments demonstrate its great application potential to construct multi-DOF manipulation devices and perform grasping operations. In short, the relevant findings and results successfully demonstrate the feasibility and superiority of the new drive mode using piezoelectric ceramic and abandoning the transmission mechanism. This work provides a new direction to develop a robotic

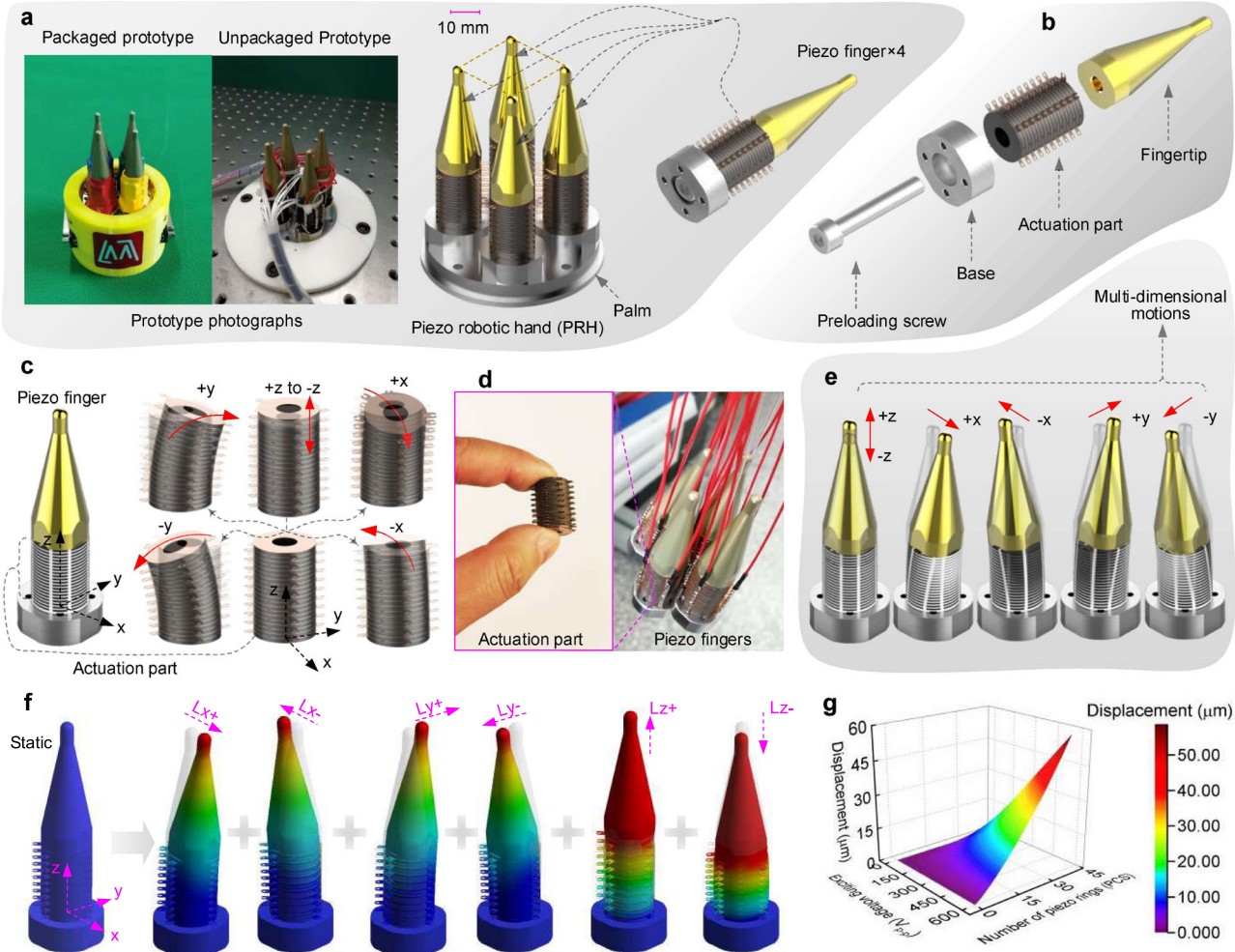

**Fig. 2 | Structural configurations of the PRH and multi-dimensional motions of the piezo finger induced by deformations of the actuation part.** **a** Prototype and overall structure of the PRH. **b** Components of single piezo finger. **c** Deformations of the actuation part, including 2D lateral bending motions along the *x*-axis and *y*-axis, as well as 1D longitudinal extending motion along the *z*-axis. **d** Photos of the actuation part and piezo fingers. **e** Multi-dimensional motions of the piezo fingers, which are induced by the deformations of the actuation part. **f** Simulated multi-dimensional motions of the piezo finger by the finite-element method. **g** Influences of the number of piezo rings and exciting voltage on the response displacement.

hand for achieving multi-DOF motion manipulation from the micro scale to the macro scale.

## Results

### Design and characterization of the PRH

For manipulating various objects to produce multi-DOF motions, we design and fabricate a unique PRH composed of four piezo fingers and a palm (Fig. 2a). The four piezo fingers are distributed at four corners of a square and fastened on the palm by screws. The tops of the piezo fingers are regarded as supporting ends to manipulate various objects (more details about the motion forms, DOFs, and supporting strategy of typical flatbed, cylindrical and spherical objects are presented in Supplementary Note 1 and Supplementary Fig. 1). The piezo fingers are designed as bolt-clamped metal–ceramic sandwich structures. In detail, each piezo finger consists of a base (designed to construct the piezo finger and provide connections to the palm), a fingertip (designed to be the manipulating end for contacting with the manipulated objects), an actuation part (mainly made of a group of piezoelectric ceramic rings and used to generate manipulation motions by transforming electrical energy into the mechanical energy), as well as a preloading screw (used to connect the base and the fingertip for clamping the actuation part between them) (Fig. 2b).

With the special structural and electrical configurations of the actuation part, it can produce bidirectional bending motions and extending motions (Fig. 2c) by only one group of piezoelectric ceramic rings (the detailed configurations and deformation principles of the actuation part are presented in Supplementary Note 2 and Supplementary Fig. 2; the real actuation part, piezo fingers, and detailed components are shown in Fig. 2d and Supplementary Fig. 2g). Thus, the whole piezo finger can achieve two-dimensional (2D) lateral bending motions and one-dimensional (1D) longitudinal extending motion (Fig. 2e). The top of the piezo finger can generate motions in the corresponding directions resultantly.

We adopt the finite-element method to simulate the motions of the piezo finger (the details about the material configurations and the key dimensions of the piezo finger are presented in Supplementary Note 3 and Supplementary Fig. 3, respectively). The simulated results (Fig. 2f) show that the piezo finger successfully realizes 2D lateral bending motions and 1D longitudinal extending motion (Supplementary Movie 1). The influences of the number of piezoelectric ceramic rings and exciting voltage on the response displacement of the piezo finger are shown in Fig. 2g (more descriptions are presented in Supplementary Note 4). Then we fabricate four piezo fingers to investigate their fundamental characteristics, and further investigate the motion manipulation ability of the whole PRH. Our experiments reflect that the

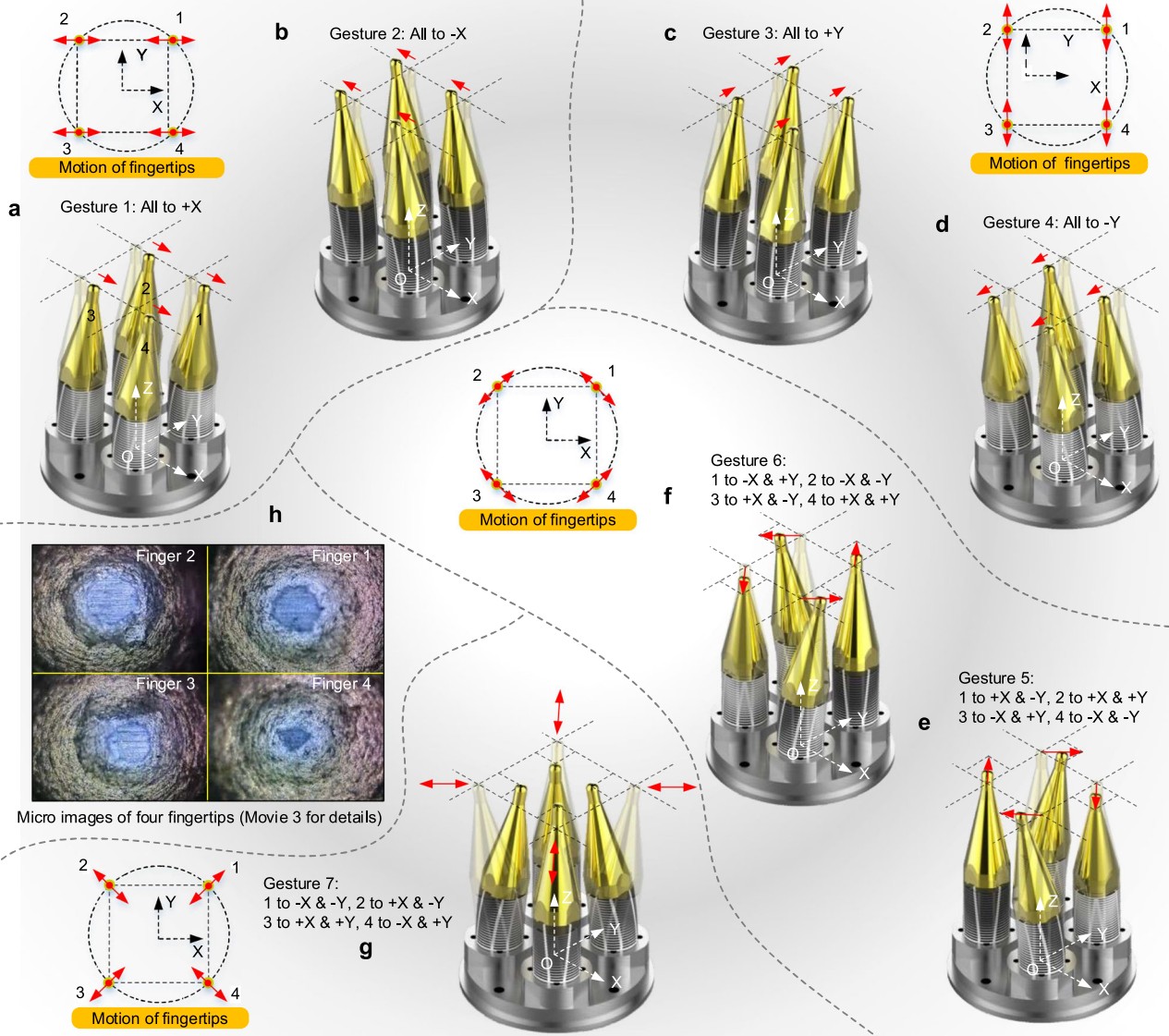

**Fig. 3 | Several typical functionalized hand gestures of the PRH, which are planned to manipulate various objects. a** and **b** show gesture 1 and gesture 2, in which all fingers bend along positive and negative directions of the *X*-axis, respectively. **c** and **d** indicate gestures 3 and 4, in which all fingers bend along positive and negative directions of the *Y*-axis, respectively. **e** and **f** represent gestures 5 and 6, in which all fingers bend along the clockwise and anticlockwise tangent direction of the circumscribed circle of their distribution position in the top view, respectively. **g** shows gesture 7 for grasping operation. **h** Micro photos of four fingertips.

piezo fingers hold features of low hysteresis (<3.95%), high resolution (about 15 nm), fast response (0.5 ms), and high natural frequency (4.1 kHz) (Supplementary Note 5 and Supplementary Fig. 4). Each piezo finger can achieve three motion dimensions, and the whole PRH can be regarded as having 12 motion DOFs.

## Functionalized hand gestures

We plan some functionalized hand gestures of the PRH by combining the multi-dimensional motions of the four piezo fingers. (i) Gestures 1 and 2: all fingers bend along the *X*-axis, meanwhile the motions of all fingertips along the *X*-axis are produced resultantly (Fig. 3a and b). (ii) Gestures 3 and 4: all fingers bend along the *Y*-axis, meanwhile, the motions of all fingertips along the *Y*-axis are produced resultantly (Fig. 3c and d). (iii) Gesture 5: finger 1 bends along the positive direction of the *X*-axis (labeled as +X) and negative direction of the *Y*-axis (labeled as −Y) simultaneously, finger 2 bends along the positive direction of *X* and *Y* axes simultaneously, finger 3 bends along negative direction of the *X*-axis (labeled as −X) and positive direction of *Y*-axis (labeled as +Y) simultaneously, and finger

4 bend along −X and −Y simultaneously, the four fingertips produce motions along the clockwise tangent direction of the circumscribed circle of their distribution positions in the top view (Fig. 3e). (iv) Gesture 6: the bending directions of all fingers are reversed exactly from gesture 5, meanwhile the four fingertips produce motions along the anticlockwise tangent direction of the circumscribed circle of their distribution positions in the top view (Fig. 3f). (v) Gesture 7: finger 1 bends along −X and −Y simultaneously, finger 2 bends along +X and −Y simultaneously, finger 3 bends along +X and +Y simultaneously, finger 4 bends along −X and +Y simultaneously (Fig. 3g). Notably, the above gestures only combine the lateral bending motions of the four piezo fingers. Moreover, we can also plan more functionalized hand gestures by using the longitudinal extending motions of the piezo fingers (gestures 8–13 shown in Supplementary Fig. 5 and Supplementary Note 6). The more intuitive animations of these functionalized gestures are presented in Supplementary Movie 2. The micro photos of the four piezo fingertips are shown in Fig. 3h. Since multi-dimensional motions can be induced on the tops of the fingertips in specific directions through

these gestures (Supplementary Movie 3), we can apply them to manipulate various objects for achieving multi-DOF motions or grasping objects.

## Manipulation mechanisms and characteristics

One goal of our research is to explore the way using the functionalized gestures of the PRH to manipulate diverse objects. Here, choosing a square plate as an example, we introduce a manipulation mechanism using the alternating control of dynamic friction and static friction to achieve multi-DOF manipulation. Then the manipulating characteristics are evaluated by a series of experiments.

The periodical saw-tooth exciting signal (Fig. 4a) can be used to excite hand gestures 1 and 2, manipulating the plate for achieving linear motion. Each period $T$ of the saw-tooth signal contains an ascending phase and a descending phase; we label the start of the ascending phase, the end of the ascending phase, and the end of the descending phase as moments 1–3, respectively. The plate can be manipulated to move a step in one period by following two phases (Fig. 4b).

Phase I (Static friction action): with the voltage increasing slowly from the minimum to the maximum in ascending phase, the gesture of the PRH changes from gesture 2 to gesture 1 slowly. The manipulating state of the PRH changes from moment 1 to moment 2. The plate is manipulated to move along $X_p$ axis by static friction action between the plate and the fingertips.

Phase II (Dynamic friction action): while the voltage drops suddenly from the maximum to the minimum in descending phase, the gesture of the PRH returns to gesture 2 from gesture 1 suddenly. The manipulating state of the PRH changes from moment 2 to moment 3. Meanwhile, the plate keeps the already moved displacement by relative sliding between the plate and the fingertip due to motion inertia.

With a similar mechanism, the plate can be moved a linear step along $Y_p$ axis by exciting the hand gestures 3 and 4 (Fig. 4c); it can also be manipulated to rotate an angular step around $Z_p$ axis by exciting the hand gestures 5 and 6 (Fig. 4d). Notably, the large range motions can be achieved by repeating excitation, and the motion directions can be reversed by reverting the exciting signal. We abstract the mathematical descriptions of the above two phases: as for phase I using static friction to slowly manipulate the plate, the inertial force or moment of the plate cannot overcome the maximum static friction between the plate and the fingertips (i.e., Eq. (1) is not satisfied), so that the plate just as sticking on the fingertips to move or rotate. While for phase II, the inertial force or moment of the plate exceeds the maximum static friction between the plate and the fingertips (i.e., Eq. (1) is satisfied), so that the relative sliding motion occurs, in which the plate maintains the already moved step due to motion inertia and the fingertips return to the initial hand gesture.

$$\begin{cases} m\frac{d^2x}{dt^2} > F_{f0,max} \\ J\frac{d^2\theta}{dt^2} > M_{f0,max} \end{cases} \quad (1)$$

where $m$ and $J$ are the mass and the rotary inertia of the plate, $x$ and $\theta$ denote the linear and rotary displacements of the plate, $F_{f0,max}$ and $M_{f0,max}$ represent the maximum static frictional force or frictional moment between the plate and the four fingertips, respectively.

Notably, with a similar manipulating mechanism, the cylindrical and spherical objects can be also manipulated to produce multi-DOF motions (Supplementary Fig. 6). The intuitive animations of manipulating the flatbed object to achieve 2L + 1R motions, manipulating the cylindrical object to achieve 1L + 1R motions and manipulating spherical object to achieve 3R motion are presented in Supplementary Movie 4.

For investigating the practical manipulating characteristics, we set the exciting signal as saw-tooth signals to excite multi-DOF motions of the square plate. Then the displacements under different exciting voltages were tested (Fig. 4e–g), and the corresponding manipulating velocities were obtained (Fig. 4h), in which the tests under one condition were repeated five times. These results indicate that the PRH successfully manipulates the plate to produce 2L + 1R motions in both forward and backward directions; the velocities of the plate are approximately linear with the voltages so a feasible method to control the velocity is to adjust the exciting voltage.

Next, we set the exciting voltage to a maximum of 600 $V_{p-p}$ and changed the exciting frequency to investigate its influence on the velocity. Then the 2L + 1R motions under diverse exciting frequencies were recorded (Fig. 4i–k). We continued to increase the exciting frequency and obtained a series of velocities under diverse frequencies (Fig. 4l). The tested results show that the velocity ascends with the increase of frequency within specific ranges, while it descends after the frequency increases to some key frequencies. Quantitatively, the highest velocities of $LX_p$ and $LY_p$ DOFs reach 5912.60 and 6000.01 µm/s at the frequency of 330 Hz, respectively, and the velocity is approximately linear with the frequency <330 Hz; the highest velocity of $RZ_p$ DOF reaches 382.47 mrad/s at the frequency of 270 Hz, and the velocity is also approximately linear with the frequency when it is <270 Hz. These results reveal that the reliable manipulating frequency ranges of linear and rotary motions of the plate are less than 330 and 270 Hz, respectively. Therefore, the velocity can also be controlled by adjusting the exciting frequency within these ranges. Notably, with the increase of the exciting frequency, the time occupied by the static friction stage in one exciting period becomes shorter and shorter, and the sliding motion between the fingertips and the manipulated object will occur when the time of this stage is shortened to a certain extent. This sliding motion results in the decrease of the displacement of the static friction action stage, which causes the decrease of the effective displacement step pitch in a single manipulation period, and the manipulation velocity decreases resultantly.

In order to investigate the manipulating characteristic with carrying a load, we applied additional metal weights on the plate (Fig. 4m). On the one hand, we set the exciting voltage and frequency as 600 $V_{p-p}$ and 1 Hz, respectively, and tested the manipulating velocities of $LX_p$, $LY_p$, and $RZ_p$ DOFs by gradually increasing the carrying weight. Then the influence of the carrying loads on the velocities was obtained (Fig. 4n). The results show that the velocities gradually decrease in a linear trend with the increase of loads. Quantitatively, the no-load velocities of $LX_p$, $LY_p$, and $RZ_p$ DOFs are 18.73, 19.74 µm/s, and 1.57 mrad/s, respectively. The tiny difference between the manipulation velocities on $X$-axis and $Y$-axis can be mainly attributed to two factors. One is the different response characteristics of a piezo finger in the $X$ and $Y$ axis, which is mainly caused by machining and assembly errors of the piezoelectric ceramic rings. Another is that the frictional characteristics between the piezo fingers and the manipulated object are not exactly the same on the $X$-axis and $Y$-axis. When the carrying load is increased to 14.76 kg, the manipulating velocities of $LX_p$, $LY_p$, and $RZ_p$ DOFs drop to 4.35, 5.75 µm/s, and 1.18 mrad/s, corresponding to 23.22%, 29.13%, and 75.16% of the no-load results, respectively. Notably, the small size of the PRH causes the inconvenience of further increasing the carrying load, so we just apply a maximum load of 14.76 kg in our test (Supplementary Movie 5). But we can predict the stuck carrying loads to about 19.22, 20.83, and 59.42 kg for $LX_p$, $LY_p$, and $RZ_p$ DOFs by fitting linear trends between the velocities and the loads, respectively. Furthermore, we also investigated the velocities under different exciting voltages with carrying loads of 0, 7.38, and 14.76 kg (Fig. 4o–q). These results show that the exciting voltage has obvious dead zones under carrying loads of 7.38 and 14.76 kg, which means that it is necessary to increase the exciting voltage for manipulating large loads. To sum up, the above experiments fully demonstrate that our PRH holds the delightful capability of manipulating large loads, achieving a tremendous ratio of 49.28 between the stuck

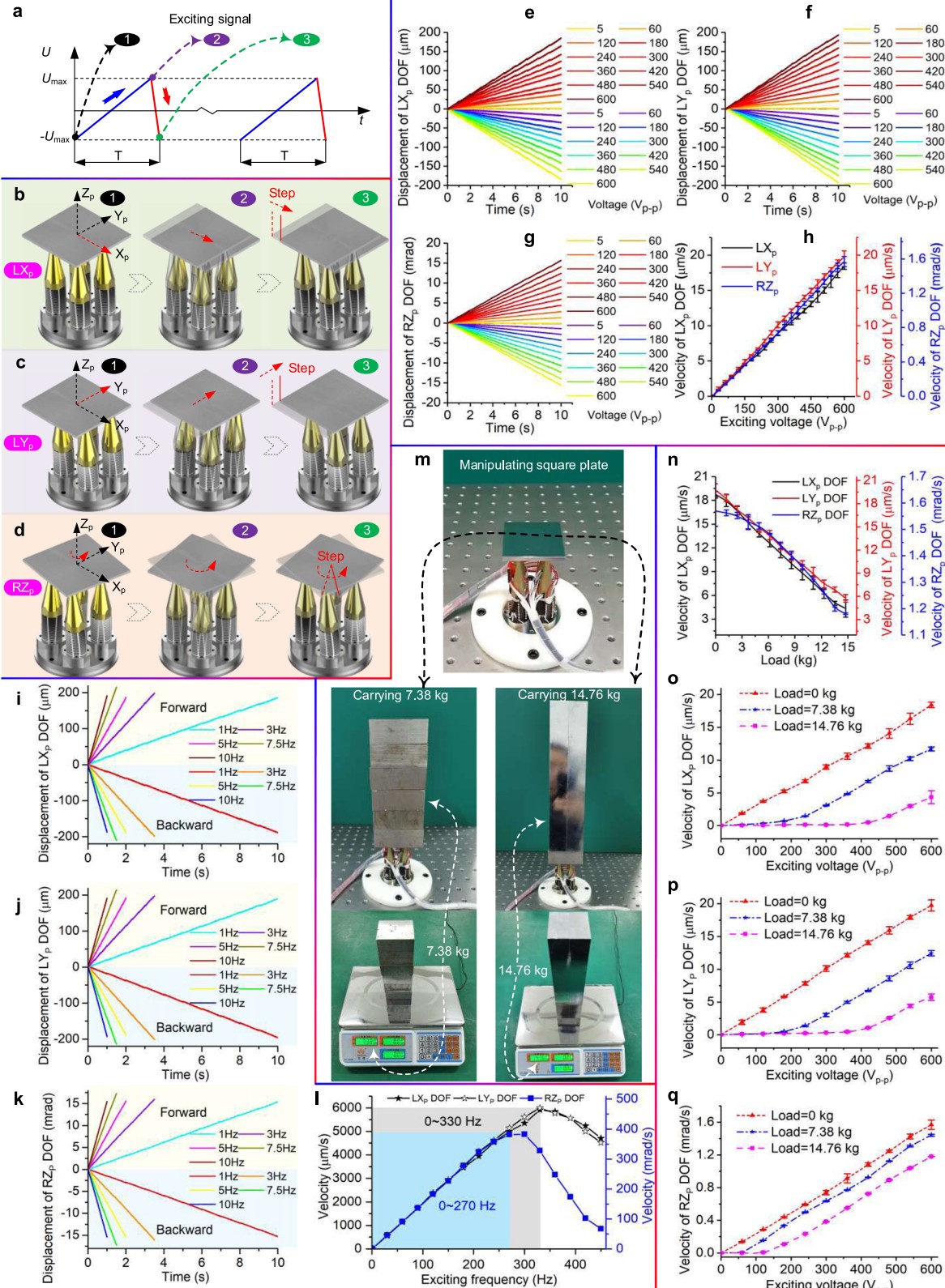

**Fig. 4 | Mechanisms and characteristics of manipulating a plate. a** Diagram of the periodical saw-tooth exciting signal. **b**–**d** Processes of manipulating plate to produce one displacement step in a single period: linear motion along $X_p$ axis (labeled as $LX_p$), linear motion along $Y_p$ axis (labeled as $LY_p$) and rotary motion around the $Z_p$ axis (labeled as $RZ_p$). **e**–**g** Motion of the plate under different exciting voltages: bidirectional motions of $LX_p$, $LY_p$, and $RZ_p$ DOFs, respectively. **h** Relationship between the manipulating velocity and the exciting voltage. **i**–**k** Motion of the plate under different frequencies: bidirectional motions of $LX_p$,

$LY_p$, and $RZ_p$ DOFs, respectively. **l** Relationship between the manipulating velocity and the exciting frequency. **m** Photos of manipulating the plate with carrying loads of 0, 7.38, and 14.76 kg. **n** Motion manipulation characteristics carrying different loads: the manipulating velocity versus carrying the load. **o**–**q** Manipulating velocity versus the exciting voltage of $LX_p$, $LY_p$, and $RZ_p$ DOFs carrying loads of 0, 7.38, and 14.76 kg, respectively. Note: the error bars represent the measurement deviation of five repeated tests.

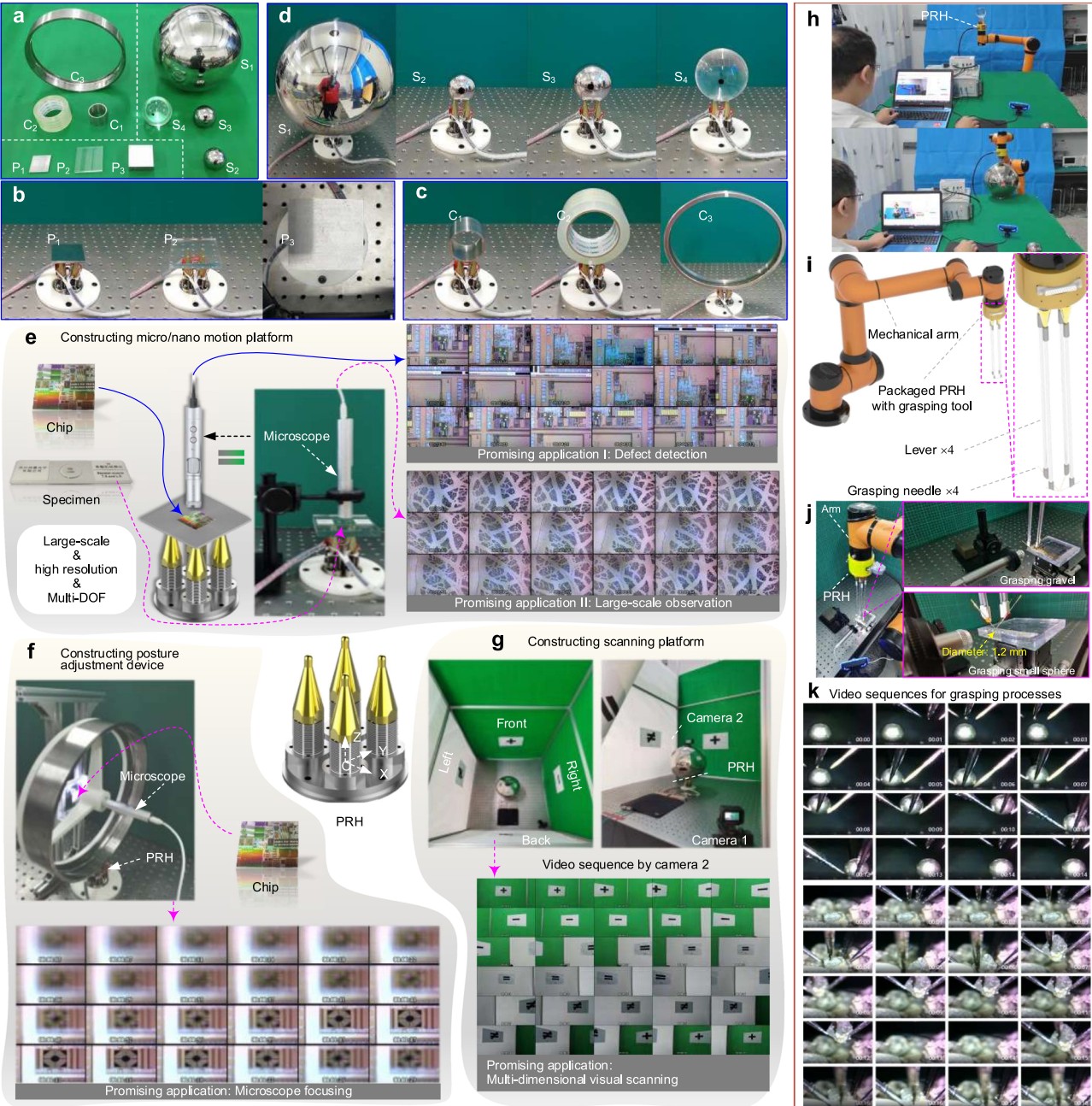

**Fig. 5 | Manipulation experiments of various objects and promising application scenarios of our PRH. a** The selected ten typical manipulated objects. **b** Photos of manipulating three flatbed plates $P_1$–$P_3$. **c** Photos of manipulating three typical cylindrical objects $C_1$–$C_3$. **d** Photos of manipulating four typical spherical objects $S_1$–$S_4$. Note: the details about shapes, materials, and dimensions of these objects are listed in Supplementary Table 2. **e** Using the PRH to manipulate plate for constructing 2L + 1R micro/nano motion platform, which can serve for defect detection in semiconductor manufacturing and micro observation in biomedical fields. **f** Using the PRH to manipulate cylindrical objects for constructing 1L + 1R posture adjustment device, which can carry the imaging device to achieve focusing. **g** Using the PRH to manipulate spherical objects for constructing 3R motion scanning platform, which can carry the camera to achieve visual scanning. Note: the complete experimental records are presented in Supplementary Movie 8. **h** The packed PRH is stalled on a 6-DOF mechanical arm to extend manipulation DOFs and implement in-situ motion manipulation. **i** Diagram of the packaged PRH installing four levers and needles on the fingertips for grasping tasks. **j** Experimental setup for grasping a small sphere and gravel. **k** Video sequences for grasping small spheres and gravel from Supplementary Movie 10.

carrying load and the self-weight (0.39 kg) of our PRH. The above-mentioned key quantitative manipulating characteristics are listed in Supplementary Table 1.

Furthermore, the noise characteristics and thermal characteristics of the PRH were also investigated. The results show that the working noise level of the PRH is within an acceptable range and there is no obvious heating phenomenon for the PRH under continuous operation (Supplementary Notes 7 and 8).

## High adaptability manipulation of various objects

Our PRH holds the superiority in manipulating various objects to achieve multi-DOF motions by virtue of multi-finger collaborations. We select 10 typical objects with diverse shapes, materials, and dimensions to conduct demonstration experiments (the flatbed objects $P_1$–$P_3$, cylindrical objects $C_1$–$C_3$, and spherical objects $S_1$–$S_4$ shown in Fig. 5a). The more details about shapes, materials, and dimensions of these objects are listed in Supplementary Table 2.

We successfully manipulate these objects to achieve multi-DOF motions; the related photos of manipulating the flatbed, cylindrical and spherical objects are shown in Fig. 5b–d, respectively. Specifically, the flatbed objects were manipulated to perform 2L + 1R motions (Supplementary Movie 6); the cylindrical objects were manipulated to achieve 1L + 1R motion, and the spherical objects were manipulated to perform 3R motions (Supplementary Movie 7). To sum up, these demonstration experiments fully show that our PRH holds high adaptability to manipulate various objects with different shapes, materials, and dimensions for achieving linear and rotary multi-DOF motions.

### Application prospects of the PRH

The above experiments show various flatbed, cylindrical and spherical objects can be effectively manipulated to achieve multi-DOF motions. The flexible motions of various objects can provide excellent foundations for constructing multi-DOF application devices. To fully demonstrate the application potentials of our PRH, the following several application scenarios were investigated.

(i)   Scenario of constructing 2L + 1R micro/nano motion platform: Benefiting from the superior characteristic of manipulating plate to achieve 2L + 1R and large-scale motions, a micro/nano motion platform was constructed (Fig. 5e). The platform carried the chips and the biological samples to achieve 3-DOF motions. A microscope was set above the platform to observe the micro details of the chips and biological samples. Then the preliminary applications of defect detection of the microcircuits in the chip and observation of the biological samples were demonstrated. More importantly, if additional precise micromanipulators, microprobes, or micromachining tools are introduced as the execution ends in the future, micro-assisted applications including manipulation, detection, machining, and assembling can be achieved promisingly. A typical example is to construct an atomic force microscope (AFM).

(ii)  Scenario of constructing 1L + 1R posture adjustment device: For demonstrating the application prospects of the PRH manipulating cylindrical objects, a ring-shaped object ($C_3$ in Fig. 5a) carrying a microscope through a trigeminal stent was used to construct a posture adjustment device (Fig. 5f). Meanwhile, the circuit chip was fixed on the vertical bracket as the observed object. The microscope was successfully carried to approach and keep away from the chip by the linear motion manipulated by the PRH, which could realize focusing to obtain clear images. This scenario briefly demonstrates the promising application of carrying imaging devices to focus local surface structures of any object. More specific focusing observations can be achieved by installing high-performance imaging devices on the ring-shaped object. Furthermore, the rotary motion can also provide the posture adjustment function for the carried imaging devices.

(iii) Scenario of constructing 3R motion scanning platform: our PRH holds the ability to manipulate the spherical object for producing 3R motions, which makes it the potential to construct 3R scanning platforms. For example, a small camera was fixed on the spherical object $S_1$ to construct a 3R visual scanning platform (Fig. 5g). This scanning platform successfully carried the camera to achieve 3R scanning motions for extending the camera view. In addition, the high rigidity and large carrying capability of our PRH would make it suit for integrating more complex devices.

The complete experimental records of the above three application scenarios are presented in Supplementary Movie 8. To further demonstrate the application value of our PRH, we packaged our PRH and specially developed a programmable power supply, a programmable gesture controller, and control software. Then an integrated PRH application system was constructed to perform application experiments. On the one hand, the PRH system was used to carry a large-scale integrated circuit chip for detect detection when worked independently; on the other hand, the integrated PRH was installed on a 6-DOF mechanical arm to extend manipulation DOF and realize in-situ motion manipulation (Fig. 5h). The corresponding experimental records are presented in Supplementary Movie 9.

In order to further demonstrate the grasping operation of the PRH, it is installed on the end of a 6-DOF mechanical arm, and four levers are installed on the four piezo fingers to extend the motion range of the fingertips. Four grasping needles are fixed on the ends of the four levers, in which the tips of the four grasping needles are adjusted to close with each other, as shown in Fig. 5i. Several small spheres (with a diameter of 1.2 mm) and a pile of gravels (size within 1.5 mm) are placed on a platform, and the PRH is used to grasp a sphere and gravel from the placed targets, as shown in Fig. 5j. The complete experimental setup is illustrated in Supplementary Fig. 8f and g. The grasping needles can be adjusted to close the target sphere or gravel by the control panel, then the target sphere is successfully grasped from one position and moved to another position by releasing with the PRH. Similarly, gravel is also grasped from a pile of gravel. The video sequences for grasping a small sphere and gravel are shown in Fig. 5k, and the complete experimental records are provided in Supplementary Movie 10. These grasping experiments fully demonstrate that the developed PRH holds great application potential for robotic grasping operations. All in all, these preliminary experiments fully demonstrate promising and significant application prospects of our PRH.

## Discussion

Our PRH exhibited a motion expansion methodology, which could promote the technical progress of multi-DOF manipulation devices including but not limited to the robotic hand. This methodology achieved the motion expansion from the micro deformations of the piezoelectric ceramic to the macro motions of various objects. Specifically, the basic actuation part made of piezoelectric ceramics was integrated into the piezo finger. It produced multi-dimensional deformations by means of the $d_{33}$ working mode. The deformations of the actuation part could induce 2D lateral motions and 1D longitudinal motion of the piezo fingers by virtue of elastic deformation. The PRH was constructed by four piezo fingers with an array configuration. Then, more than 10 functionalized gestures of the PRH were planned to generate flexible multi-dimensional motion trajectories, which were used to manipulate various objects by active alternating control of dynamic and static frictions. The PRH successfully achieved adaptive and flexible manipulations of ten objects and meanwhile exhibited features containing multi-DOF, multi-form, and large carrying load, which brought promising application prospects. The key to realizing the above motion expansion lies in two factors: (i) the generation of the multi-dimensional actuation trajectory by accumulating micro deformations and (ii) the friction coupling between the fingertips and the manipulated objects. As long as these two factors are realized through special configuration creation and method planning, the motion expansion from the micro deformations to the macro motions can be achieved successfully. This methodology provides referential and instructive ideas for creating new multi-DOF precision manipulating devices with functional piezoelectric ceramic.

With the integration of rigid structure and functional material, our PRH inherited some advantages of piezoelectric ceramic. The experimental results showed that our PRH held multi-dimensional motion ability (12 motion DOFs) and fundamental characteristics including high linearity (hysteresis <3.95%), high resolution (about 15 nm), fast response (0.5 ms), and high natural frequency (4.1 kHz). Our PRH had the capability of manipulating various objects or grasping objects with different shapes, materials, and dimensions. A stainless plate was chosen as a manipulation case, and the 2L + 1R motions were achieved successfully; the manipulating experiments showed

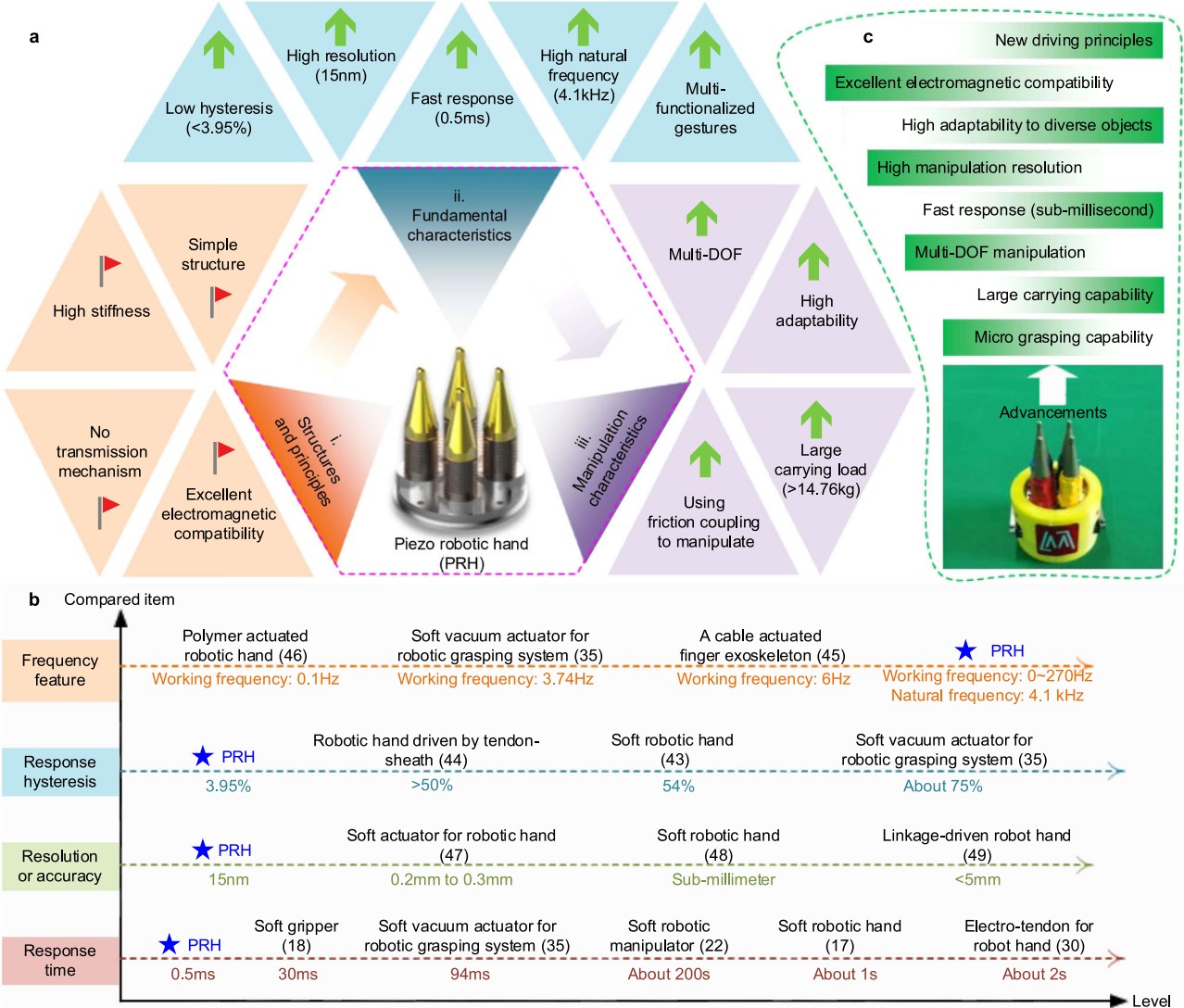

**Fig. 6 | Characteristic summary and comparison of the PRH. a** Characteristics of the PRH summarized from three aspects: (i) structures and principles; (ii) fundamental characteristics; (iii) manipulation characteristics. **b** Quantitative characteristic comparison of the PRH and other robotic hands. **c** Main advancements of this work compared with other robotic hands.

that the 3-DOF maximum velocities reached 5912.60, 6000.01 μm/s, and 382.47 mrad/s, respectively; the velocities with carrying a load of 14.76 kg only dropped to 23.22%, 29.13%, and 75.16% of the no-load ones, and the completely stuck carrying loads for the 3-DOF were estimated to be 19.22, 20.83, and 59.42 kg, respectively. The ratio between the carrying load and the self-weight (0.39 kg) reached 49.28. A series of cylindrical and spherical objects were also successfully manipulated to produce multi-DOF motions. These results fully demonstrated that our PRH held the advantages of multi-DOF, high adaptability, and large load manipulation ability, which gave it promising application potential. The above superior characteristics are just benefited from the unique design of our PRH, including integrating functional piezoelectric ceramic, adopting high stiffness structure, and abandoning transmission mechanism.

Our PRH adopts piezoelectric ceramic rings to achieve motion manipulation, the structure is designed to be rigid and without a transmission mechanism, which is obviously different from other robotic hands. The characteristics of our PRH can be concluded from aspects of structures and principles, fundamental and manipulation characteristics (Fig. 6a). In order to evaluate the characteristic level of our PRH, a quantitative comparison of the

response time, resolution, or accuracy, response hysteresis, and frequency feature of our PRH and other robotic hands is accomplished (Fig. 6b)[43–49]. It can be found that the response time of the PRH is obviously short than other robotic hands due to the fast response capability of the functional piezoelectric ceramics. Our PRH achieves excellent open-loop motion resolution of about 15 nm, which is difficult for other robotic hands. With the inherent fast electromechanical conversion of the piezoelectric ceramics, our PRH achieves low hysteresis of 3.95% obviously lower than other robotic hands. Furthermore, owing to the unique design of high rigidity and no transmission structures, our PRH achieves a high natural frequency of about 4.1 kHz; this provides a good bandwidth base for high working frequency, which is superior to other low-rigidity robotic hands. In short, our PRH holds merits including low hysteresis (<3.95%), high resolution (15 nm), fast response (0.5 ms), and high natural frequency (4.1 kHz). Owing to the unique features of the driving principle, direct-drive, and rigid structure, our PRH achieves several technical advancements (Fig. 6c): (i) our PRH holds excellent electromagnetic compatibility by using piezoelectric ceramic with new principle; (ii) the high stiffness structure without transmission mechanism and the hinge are simpler, which can avoid

the accuracy loss caused by transmission mechanisms; (iii) our PRH achieves excellent manipulating characteristics including high adaptability (benefiting from the array configuration and the diverse functionalized hand gestures) and large carrying load (benefiting from high stiffness structures); (iv) our PRH holds grasping capability for micro-objects. In summary, our PRH holds favorable technical advancements in the aspects of driving principles, structural design, and manipulating characteristics compared with the other robotic hands. Furthermore, the plate manipulated with our PRH is similar to precision stages to some extent. Thus, we briefly compare it with some typical precision stages to show its features (Supplementary Note 9 and Supplementary Table 3).

It is worth noting that we only investigate the manipulating actions by the tops of the fingertips in this work. In fact, the sides of fingertips can also be used to manipulate other objects, and specific manipulating tools can also be installed in the fingertips to achieve manipulation (Supplementary Fig. 7). These are also worthy of studying contents for our follow-up works. This work utilizes active alternating control of dynamic and static frictions to manipulate various objects for achieving macro motions, and we can also only use static friction to manipulate objects for achieving micro motions by using the planned hand gestures. We will continue to deeply study the motion manipulation principles and characteristics of our PRH.

In order to apply our research results in the future, including but not limited to the following steps need to be studied: (i) exploring more effective manipulation mechanisms to manipulate diverse objects for broadening the manipulation capability; (ii) integrating construction of robot-assisted manipulation devices using our PRH; (iii) establishing motion detection methods and studying practical realization to face specific applications; (iv) finding effective control strategy to improve manipulation characteristics.

## Methods

The components of our PRH consist of the palm, finger bases, fingertips, common electrodes, fan-shaped electrodes, piezoelectric ceramic rings, and screws (Fig. 2b and Supplementary Fig. 2a). The palm and the finger bases were fabricated with stainless steel; the fingertips were made of aluminum alloy; the common electrodes and the fan-shaped electrodes were fabricated with beryllium bronze; all the screws were standardized component made of stainless steel; all piezoelectric ceramic rings were made of PZT-4 (Lead Zirconate Titanate). The palm, the finger base, and the fingertips were fabricated by computer numerical control (CNC) machining tool, and the common electrodes and the fan-shaped electrodes were fabricated by wire electrical discharge machining (WEDM) machining tool, they were provided by Shenzhen Huiwen Smart Technology Co., Ltd., China. The piezoelectric ceramics were provided by the 46th Research Institute of China Electronics Technology Group Corporation.

The PRH prototype was assembled with five steps: (i) all components were wiped with alcohol to remove surface oil stain and impurity; (ii) according to the component configurations in Supplementary Fig. 2a, all piezoelectric ceramic rings, and electrode slices were bonded together with epoxy resin adhesive to constitute the whole actuation part, then it was solidified through aging treatment of 48 h; (iii) the finger base, the actuation part, and the fingertip were assembled together by a preloading screw to constitute the piezo finger; (iv) the copper core wires were welded to the corresponding electrodes of the four piezo fingers; and (v) the four piezo fingers were installed on the palm through some screws to constitute the whole PRH.

An xPC system was used to generate exciting signals and process the acquired signals from the displacement sensors (Supplementary Note 10 and Supplementary Fig. 8a), in which the sampling frequency was set as 10 kHz. A Doppler laser vibration system (Model: PSV-400-M2, Polytec, Germany) was used to test the amplitude-frequency characteristics of the piezo fingers. Several sensor configuration relationships were used to measure the multi-DOF output displacements of the plate (Supplementary Fig. 8b). The miniature microscope (Model: B011, Shenzhen Supereyes Co., Ltd., China) used in the experiments with amplifying factor of 1000 times. The small video camera in the third application scenario was with 4k resolution and a frame rate of 30 fps (Model: YDXJ01FM, Beijing FIMI Technology Co., Ltd., China). The videos and photos of manipulating the 10 objects, the first two application scenarios, and the experimental records of the integrated PRH system were acquired through a phone camera (Model: Mate 30 Pro 5G, Huawei, China). A digital noise meter (Model: AR824, provided by Smart senor, China) was used to test noise characteristics (Supplementary Fig. 8c). A thermal imager (Model: UTI380, UNI-Trend Technology, China) was used to test the temperature of the PRH (Supplementary Fig. 8d). The experimental setup shown in Supplementary Fig. 8e was used to implement demonstrating experiments for installing the PRH on the end of a 6-DOF mechanical arm to extend manipulation DOF and realize in-situ manipulation. The experimental configurations shown in Supplementary Fig. 8f and g were used to carry out grasping experiments.

## Data availability
All data generated or analyzed during this study are included in this published article (and its supplementary information files).

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

## Acknowledgements

This work was supported in part by the National Natural Science Foundation of China grant Nos. 52225501 (Y.L.), U1913215 (Y.L.), and 5210051275 (J.D.) and in part by Interdisciplinary Research Foundation of HIT grant No. IR2021233 (Y.L.).

## Author contributions

S.Z. and Y.L. coordinated and wrote the manuscript; Conceptualization: S.Z. and Y.L.; Methodology: S.Z., Y.L., and J.D.; Investigation: S.Z., J.D., X.G., J.L., W.W., and M.X.; Data acquisition: S.Z.; Visualization: S.Z., Y.L., and J.L.; Funding acquisition: Y.L. and J.D.; Project administration: Y.L.; Supervision: Y.L. and J.D.; Writing—original draft: S. Z., Y.L., J.D., X.G., J.L., W.W., M.X., X.M., and Q.C.; Writing—review & editing: S.Z., Y.L., J.D., J.K.L., W.C., and J.Z.

## Competing interests

The authors declare no competing interests.
