## [Peer Review File · Nature Communications]

REVIEWER COMMENTS

Reviewer #1 (Remarks to the Author):

Comments:

1. the main novelties of the paper should be summarized with bullets, especially commenting on the state of the art solutions;
2. would it be possible to test the solution in a real experiment (e.g., robotic grasping or assembly of a component)?
3. check the English.

Reviewer #2 (Remarks to the Author):

Comments to the authors

=====

In the manuscript, the authors proposed a robotic hand consisting of four piezo-based fingers that can convert the motion from micro scale to macro scale. A robotic hand with multiple degrees of freedom is a versatile and exciting platform, and the authors described a variety of application possibilities. This is a new concept; the study is data-rich, the literature research is sufficient, and the manuscript is easy to follow and understand. I recommend the authors address the comments before publication.

Comments:

- 1) "Our PRH exhibits several typical features:... (iii) realizing characteristics including high resolution, low hysteresis, fast response and high natural frequency" Quantitative comparisons with other robotic arms will be more convincing.

2) Since the working frequency of the piezo finger is in the range of human hearing, is there any noise during running?

3) The excitation voltage is high. Will the temperature in and around the finger be changed a lot? Will the change affect the performance of the finger in terms of response speed and degree? Will the temperature change affect the sample operated?

4) What is the duty of the finger? How many cycles can it work?

5) Will different amounts of piezo rings cause different response speeds and degrees? If so, give the plot.

7) "With the special structural and electrical configurations of the actuation part, it can produce bidirectional bending motions and extending motions." What is the different electrical configuration? At least show the entire details of one deformation type.

8) "The tested results show that the velocity ascends with the increase of frequency within specific ranges, while it descends after the frequency increases to some key frequencies." Why? What is the key frequency?

9) "the no-load velocities of LXp, LYp, and RZp DOFs are 18.73 $\mu\text{m/s}$, 19.74 $\mu\text{m/s}$ " why is the velocity different since the ring is symmetrical?

10) The finger design and working mechanism have been reported in the publication, Bioinspired Multilegged Piezoelectric Robot: The Design Philosophy Aiming at High-Performance Micromanipulation. Please describe in detail the novelty of this manuscript.

12) In fig.2, a scale bar may be helpful to show the size of the hand.

Reviewer #3 (Remarks to the Author):

This paper presents the development and utility of a linearly moving, highly rigid piezo-robot hand (PRH) made of functional piezoelectric ceramics.

The paper shows that the four piezoelectric fingers enable 12-DOF motion, which is difficult to achieve with existing actuators. Notably, they also show in a movie that 18 motions and 3R motions are generated by the motion of a sphere, including plate manipulation that realizes 2L (linear) and 1R (rotational) motions and cylindrical object manipulation that realizes 1L and 1R motions, and report the world's first multi-degree-of-freedom ultra-precision manipulation device.

Actuators are positioned as extremely important devices along with sensors in the next generation digital society. However, compared to sensors, research and development of actuators is more difficult and progress has been slow. In particular, with actuators, there is a trade-off between response speed, force generated, amount of displacement generated, and drive voltage. Therefore, electrical operation has limited applications, requiring complex mechanical structures such as pneumatic, hydraulic, or cable actuation.

The authors' proposal is electrically operable and resistant to electromagnetic noise, enabling extremely delicate operation, and has an extremely high generating force, which is a characteristic of piezoelectric ceramics.

The video demonstration shows the operation of very large and heavy objects as well as lightweight and cylindrical objects, and the range of applications is wide. The detailed analysis of the mechanisms and numerical analysis related to the operation is also provided, which is of great value not only in the applied aspect but also in the academic aspect.

I believe that publication in Nature Communications is appropriate after answering the following minor questions. 1.

1. The experimental results and videos are excellent and show that the behavior described in the paper is faithfully reproduced. On the other hand, there are many illustrations, etc., and almost no photos of actual device structures and components. Actual photographs should be shown along with the illustrations. In particular, it would be academically preferable for Figures 2 and 3 to be shown together with photographs rather than illustrations.

2. I believe that the object to be compared should be a precision stage, not a robotic hand. In fact, researchers studying robotic hands are aiming at flexible mechanisms, force generation,

displacement control, etc. for softly grasping eggs and other objects without breaking them, and not at the movements shown in the movies in this paper and others. The performance comparisons shown in Figure 6 and elsewhere are not accurate. If the proposed actuator is to be compared to a robotic hand, a demonstration video should be shown for comparison, such as "grabbing things".

The piezoelectric device proposed in this paper is an excellent actuator, and the detailed description of the device structure is valuable both academically and in terms of application. On the other hand, I do not think it is appropriate in comparison with a robotic hand. If a comparison with a robotic hand is to be made, such a demonstration or the like should be shown, but at present it is only used as a "precision stage".

Responses to reviewers' comments (Reviewer: #1)

Response: We sincerely thank you for your efforts in reviewing this manuscript. The manuscript has been carefully revised according to your valuable suggestions. All of the modifications have been highlighted with yellow tool in the revised manuscript and supplementary information. All of your professional comments are seriously considered and replied in detail as follows.

Comment 1. The main novelties of the paper should be summarized with bullets, especially commenting on the state-of-the-art solutions.

Response: According to your professional suggestions, we have carefully condensed and summarized the contributions of this paper. The main novelties and contributions can be concluded with bullets as follows:

- (1) This work proposes the first robotic hand with four fingers constructed on piezoelectric ceramics, it achieves in-hand multi-DOF manipulation, as well as high resolution of 15 nm, fast response of 0.5 ms, low hysteresis of 3.95% and no electromagnetic interference.
- (2) The PRH is designed to be a unique rigid configuration with four bolt-clamped metal-ceramic sandwich piezo fingers, which achieves compact structures and large load ability as it has no transmission mechanisms and hinges.
- (3) The PRH achieves motion expansion from the micro deformations of piezoelectric ceramics to the macro motions of various objects by four-finger cooperative manipulations, and realizes multi-DOF and cross-scale motion manipulation.
- (4) The PRH realizes excellent adaptability to manipulate various objects with diverse shapes, materials, and dimensions. A series of experiments demonstrate its great application potentials to construct multi-DOF manipulation devices and perform grasping operations.

Modification: The above contents have been added into the last paragraph of the Introduction section in the revised manuscript as follows:

“...The main novelties and contributions of this work can be concluded as: (i) This work proposes the first robotic hand with four fingers constructed on piezoelectric ceramics, it achieves in-hand multi-DOF manipulation, as well as high resolution of 15 nm, fast response of 0.5 ms, low hysteresis of 3.95% and no electromagnetic interference. (ii) The PRH is designed to be a unique rigid configuration with four bolt-clamped metal-ceramic sandwich piezo fingers, which achieves compact structures and large load ability as it has no transmission mechanisms and hinges. (iii) The PRH achieves motion expansion from the micro deformations of piezoelectric ceramics to the macro motions of various objects by four-finger cooperative manipulations, and realizes multi-DOF and cross-scale motion manipulation. (iv) The PRH realizes excellent adaptability to manipulate various objects with diverse shapes, materials, and dimensions. A series of experiments demonstrate its great application potentials to construct multi-DOF manipulation devices and perform grasping operations.”

Comment 2. Would it be possible to test the solution in a real experiment (e.g., robotic grasping or assembly of a component)?

Response: Your concern about more demonstration experiments is important and instructive. As for the developed piezo robotic hand, it can not only manipulate various objects to achieve multi-DOF motions with the preplanned functionalized gestures, but also be used to grasp some objects to realize transfer or other operations. The former has been demonstrated with a series of experiments, as shown in the initial

submission. According to your professional suggestions, a grasping application experiment of the piezo robotic hand has been planned and carried out. We believe that these real grasping experiments can prove the application value and service capability of the developed piezo robotic hand.

As mentioned in the previous manuscript, more functionalized gestures can be obtained by using motion combinations of the four piezo fingers. Thus, a functionalized gesture for grasping operations has been planned and added into the revised manuscript, as shown in the revised Fig. 3g. The packaged prototype of the piezo robotic hand can be installed on the end of a 6-DOF mechanical arm to achieve function extension. For example, extending manipulation DOF of the mechanical arm and manipulating object for in-situ motion, as the experimental records shown in Supplementary Movie 9 and the revised Fig. 5h. In order to further demonstrate the grasping operation of the piezo robotic hand, as shown in revised Fig.5i, it is installed on the end of the mechanical arm, and four levers are installed on the four piezo fingers to extend the motion range of the fingertips. Four grasping needles are fixed on the ends of the four levers, in which the tips of the four grasping needles are adjusted to close with each other. Several small sphere (with diameter of 1.2 mm) and a pile of gravels (size within 1.5 mm) are placed on a platform, and the piezo robotic hand is used to grasp a sphere and a gravel from some of them, as shown in the revised Fig. 5j. The complete experimental setup is illustrated in the revised Supplementary Fig. 8f and Fig. 8g. The grasping needles can be adjusted to close the target sphere or gravel by the control panel, then the target sphere is successfully grasped from one position and moved to another position by releasing with the piezo robotic hand. Similarly, a gravel is also grasped from a pile of gravels. The video sequences for grasping a small sphere and a gravel are shown in the revised Fig. 5k, and the corresponding experimental records are provided in Supplementary Movie 10. These grasping experiments fully demonstrate that the developed piezo robotic hand holds great application potentials for robotic grasping operations, which we hope meet with your approval. In the future, we will focus on deep researches in grasping structures, strategies, control methods, etc.

Modification: The grasping functionalized gesture of the piezo robotic hand has been added into Fig. 3g. The experimental configurations and results of the real grasping application have been added into the revised manuscript as follows.

“...In order to further demonstrate the grasping operation of the PRH, it is installed on the end of a 6-DOF mechanical arm, and four levers are installed on the four piezo fingers to extend the motion range of the fingertips. Four grasping needles are fixed on the ends of the four levers, in which the tips of the four grasping needles are adjusted to close with each other, as shown in Fig. 5i. Several small spheres (with diameter of 1.2 mm) and a pile of gravels (size within 1.5 mm) are placed on a platform, and the PRH is used to grasp a sphere and a gravel from the placed targets, as shown in Fig. 5j. The complete experimental setup is illustrated in Supplementary Fig. 8f and Fig. 8g. The grasping needles can be adjusted to close the target sphere or gravel by the control panel, then the target sphere is successfully grasped from one position and moved to another position by releasing with the PRH. Similarly, a gravel is also grasped from a pile of gravels. The video sequences for grasping a small sphere and a gravel are shown in Fig. 5k, and the complete experimental records are provided in Supplementary Movie 10. These grasping experiments fully demonstrate that the developed PRH holds great application potentials for robotic grasping operations.”

Revised Fig. 5 in the revised manuscript (See i, j and k to know the configurations and results of grasping experiments)

Comment 3. Check the English.

Response: Thanks a lot for your reminder to improve the grammar and expression. All details of the manuscript and supplementary materials have been carefully checked by the authors. We cautiously believe that the expressions in this article are already improved significantly to avoid grammar errors or inappropriate expressions.

Responses to reviewers' comments (Reviewer: #2)

Comments to the Author:

In the manuscript, the authors proposed a robotic hand consisting of four piezo-based fingers that can convert the motion from micro scale to macro scale. A robotic hand with multiple degrees of freedom is a versatile and exciting platform, and the authors described a variety of application possibilities. This is a new concept; the study is data-rich, the literature research is sufficient, and the manuscript is easy to follow and understand. I recommend the authors address the comments before publication.

Response: We sincerely thank you for your efforts in reviewing this article, your approval and valuable suggestions. Your professional comments are very helpful for the improvement of this article and the direction of our follow-up work. The manuscript has been carefully revised according to your suggestions. All of the modifications have been highlighted with yellow tool in the revised manuscript and supplementary information. In addition, we have carefully replied to your comments point-to-point as follows, which we hope meet with your approval again.

Comment 1. “Our PRH exhibits several typical features:... (iii) realizing characteristics including high resolution, low hysteresis, fast response and high natural frequency” Quantitative comparisons with other robotic arms will be more convincing.

Response: It is necessary to explain the meanings of these parameters. Resolution represents the minimum achievable and stable displacement of robotic hand, which is limited by the precision of the actuation components. Hysteresis indicates the proportion of lags between forward and reverse actions, which relies on the response speed of the forward and reverse actions. Fast response means that the time for generating full response is very short, which depends on the response speed of the actuation components and principles. The natural frequency indicates the first resonant frequency of the robotic hand, which limits the working frequency range and reflects the stiffness of the robotic hand.

In fact, we claim the statement of exhibiting features including high resolution, low hysteresis, fast response and high nature frequency after qualitative analysis in the aspects of principles, structures and transmission mechanisms of other various robotic hands. Our previous qualitative analyses are as follows:

- (1) About the motion resolution: lots of other robotic hands utilize electromagnetic motor, artificial muscle and soft actuator as the core actuation components, and the motion resolution of the whole robotic hand depends on the motion precision of the actuation components. Generally, the electromagnetic motor (servo motor, stepper motor, etc.) is an actuation component with relatively high precision, but its motion precision depends on the subdivision excitation. The motion resolution of the robotic hands based on the electromagnetic motor is usually within micron level after motion conversion with transmission mechanisms. In contrast, the tested results show the motion resolution of the developed piezo robotic hand is about 15 nm. Therefore, we believe that our piezo robotic hand holds feature of high resolution.
- (2) About the hysteresis: some other robotic hands use artificial muscle or soft actuator as the actuation components. Owing to the limits of the energy and motion conversion principles, these robotic hands have obvious lags between their forward response and reverse response. The reported results reach 50% or even more for tendon-actuated robotic hand and soft robotic hand. In contrast, the output hysteresis of our piezo robotic hand is tested as 3.95%. Therefore, we believe that our piezo robotic hand holds feature of low hysteresis.

- (3) About response time: the response time of robotic hands mainly depends on the response time of the actuation components and the motion conversion, especially for the former. Lots of reports show that other robotic hands using soft actuator usually hold slow response time due to the material's softness and structural compliance (more than tens of milliseconds or even tens of seconds). In contrast, the response time of our piezo robotic hand is tested as about 0.5 ms, which is obviously fast than other robotic hands. Therefore, we believe that our piezo robotic hand holds feature of fast response.
- (4) As for the natural frequency: lots of other robotic hands hold jointed structure or soft structure, these structural features limit their resonant frequency. In contrast, our piezo robotic hand holds unique design of high stiffness and compact structure, which brings the feature of relatively high natural frequency. Therefore, we believe that our piezo robotic hand holds feature of high natural frequency.

According to your constructive suggestions, we have carried out more deep literature investigations to find out the reported results about resolution, hysteresis, response time and nature frequency of different robotic hands. Then, a brief quantitative characteristic comparison between our work and other robotic hands has been performed. The corresponding contents have been added into the revised manuscript.

Modification: According to your suggestion, the quantitative characteristic comparison among our work and other robotic hands has been added into the revised manuscript as follows.

Fig. 6. Characteristic summary and comparison of the PRH. a Characteristics of the PRH summarized from three aspects: (i) structures and principles; (ii) fundamental characteristics; (iii) manipulation characteristics. **b** Performance comparison of the PRH and other robotic hands. **c** Main advancements of this work compared with other robotic hands.”

“...The characteristics of our PRH can be concluded from aspects of structures and principles, fundamental and manipulation characteristics (see Fig. 6a). In order to evaluate the characteristic level of our PRH, a quantitative comparison about the response time, resolution or accuracy, response hysteresis, and frequency feature of our PRH and other robotic hands is accomplished (Fig. 6b). It can be found that the response time of the PRH is obvious short than other robotic hands due to the fast response capability of the functional piezoelectric ceramics. Our PRH achieves excellent open-loop motion resolution of about 15 nm, which is difficult for other robotic hands. With the inherent fast electromechanical conversion of the piezoelectric ceramics, our PRH achieves low hysteresis of 3.95% that obviously lower than other robotic hands. Furthermore, owing to the unique design of high rigidity and no transmission structures, our PRH achieves high natural frequency of about 4.1 kHz; this provides a good bandwidth base for high working frequency, which is superior to other low-rigidity robotic hands. In short, our PRH holds merits including low hysteresis (<3.95%), high resolution (15 nm), fast response (0.5ms) and high natural frequency (4.1kHz).”

Comment 2. Since the working frequency of the piezo finger is in the range of human hearing, is there any noise during running?

Response: According to the results of motion manipulation experiments, the working frequency of our piezo robotic hand can be set within ranges of 0 to 270 Hz (Considering the linear relationship ranges). In this range, there are indeed some sounds when working. In order to evaluate the noise level during operation, we carry out a noise experiment under different working frequency by a digital noise meter (Model: AR824, provided by Smart sensor, <http://www.smartsensor.cn>), in which the working voltage was set as the maximum of 600 V_{p-p}. The experimental setup and tested results are shown in the following Fig. R2-1, in which the noise meter is configured with distance of 100 mm away from the piezo robotic hand and the noise level under each working frequency is recorded five times to reduce the random measurement error. The tested results show that the noise level increases slightly with the increase of the working frequency; the noise level at the maximum working voltage of 600 V_{p-p} and the maximum frequency of 270 Hz are less than 53.74 dB. This noise level is acceptable and far lower than the endurable limit of human ear. The noise level under higher frequency up to 360 Hz is also tested, which is also acceptable.

It is worth noting that more suitable working condition of our piezo robotic hand is intermittent high-precision motion manipulation or grasp manipulation, and the working frequency of our piezo robotic hand can be set as lower level (for example, below the lower limit of the hearing range of human ear), which enables us to maintain the noise level within the comfortable range.

Fig. R2-1 Experimental setup and results of noise characteristics. **a** Experimental configuration. **b** Tested results of noise level under different working frequency, in which the error bars represent absolute deviation of five times measurement.

Modification: A brief discussion about the noise experiment has been added into the supplementary information with the revised manuscript, please see Supplementary Fig. 4g, Fig.8c and Supplementary Note 7 as follows:

“Supplementary Note 7

Description for noise experiments of the PRH

Good man-machine compatibility is a factor worth considering, but the working frequency of the PRH at hundreds of Hertz may produce working noise due to the fast excitation of the piezoelectric ceramic components. Thus, we carried out noise experiment of the PRH under maximum working voltage of 600 V_{p-p} and different working frequencies, in which a digital noise meter is used to capture the noise level of the PRH when working (Supplementary Fig. 8c). The tested results show that the noise level of the PRH is less than 53.74 dB under maximum working frequency of 270 Hz and less than 57.42 dB under the maximum tested frequency of 360 Hz (Supplementary Fig. 4g). The working noise 53.74 dB under the maximum working frequency of 270 Hz is within acceptable level of human ear, just like the noise level of loud talking (far away from the general endurable limit 100 dB of human ear). It is worth noting that the potential working scenario of our PRH is intermittent high-precision motion manipulation, and its working frequency can be set as low level, which enables us to maintain the noise level within the comfortable range.”

Comment 3. The excitation voltage is high. Will the temperature in and around the finger be changed a lot? Will the change affect the performance of the finger in terms of response speed and degree? Will the temperature change affect the sample operated?

Response: The influence of the working time on the temperature of our piezo robotic hand (PRH) is indeed of great concern. In general, the heating phenomenon of piezoelectric devices mainly comes from heat loss of the piezoelectric ceramic components, especially for ultrasonic frequency working scenario like ultrasonic piezoelectric transducers. High power and high frequency excitations are two main factors. However, our PRH works with relatively low frequency range of 0~270 Hz, which is helpful to avoid the heating problem. Although the maximum excitation voltage reaches 600 V_{p-p} , the PRH only utilizes a power supply with peak output power of 24 W, which means that the excitation current of the power supply is too small to induce serious heating influence on the piezo finger (low current means low heat loss). It is obviously different with most of ultrasonic piezoelectric transducers requiring excitation devices with output power of hundreds of Watts or more. In summary, owing to the working condition of low power and relatively low working frequency, our PRH holds characteristic of low power consumption. Furthermore, the integrated structure of the piezoelectric finger has a considerable surface area, which also provides sufficient heat dissipation conditions for heating release.

In order to evaluate the heating level of the PRH, we carried out a thermal characteristic experiment. As shown in Fig. R2-2a, the thermal imager (Model: UTI380, UNI-Trend Technology Co., Ltd, China) and the laser displacement sensor (Model: LK-H020, Keyence Co. Ltd, Japan) are utilized to capture the thermal image and response displacement of the PRH. The PRH is continuously excited with the maximum working voltage of 600 V_{p-p} and the maximum working frequency of 270 Hz for more than 60 min, and the temperature and the response displacement are measured by time interval of 5 min. The tested results show that the temperature of the piezo finger (part of piezoelectric rings and part of fingertip) keeps no obvious change, and the response displacement on the fingertip is also unchanged, as shown in Fig. R2-2b and Fig. R2-2c. The thermal images of the PRH under different working time are shown in Fig. R2-2d to Fig. R2-2g. These experiments reflect that our PRH has no heating phenomenon when working continuously with the maximum working voltage and frequency. The temperature around the piezo finger

is unchanged when working, which means that the response characteristics including the response speed and amplitude of the piezo finger will not be affected by the nonexistent heating problem. Thus, the operated samples of the PRH are also not affected.

Fig. R2-2 Experimental setup and results of thermal characteristics. **a** Experimental configuration. **b** Temperature of the piezoelectric rings and fingertip of finger 1 under working time of 0 min to 60 min, in which the error bars represent the absolute deviation under five times measurement. **c** The response displacements under different working time. **d**, **e**, **f**, and **g** show thermal images of the PRH when working for 0 min, 20 min, 40 min and 60 min, respectively. Note: the hottest parts in the thermal images are laser head.

Modification: The following contents about the thermal characteristic experiments have been added into the supplementary information with the revised manuscript.

“Supplementary Note 8

Description for thermal characteristic experiments

In order to evaluate the heating level of the PRH when working, we carried out a thermal characteristic experiment. A thermal imager (Model: UTI380, UNI-Trend Technology (China) Co., Ltd, China) and a laser displacement sensor (Model: LK-H020, Keyence Co. Ltd, Japan) are utilized to capture the thermal image and response displacement of the PRH (Supplementary Fig. 8d). The PRH is continuously excited with the maximum working voltage of 600 V_{p-p} and the maximum working frequency of 270 Hz for more than 60 min, and the temperature and the response displacement are measured by time interval of 5 min. The tested results show that the temperature of the piezo finger (part of piezoelectric rings and part of fingertip) keeps no obvious change with continuous working time of 60 min, and the response displacement on the fingertip is also unchanged (Supplementary Fig. 4h and Fig. 4i). The thermal images of the PRH under different working time are shown in Supplementary Fig. 4j to Fig. 4m. These experiments reflect that our PRH has no heating phenomenon when working continuously with the maximum working voltage and frequency. This helps to ensure the stability of its own characteristics and avoid the influence of the PRH on the manipulated objects. The temperature around the piezo finger is unchanged when working, which means that the response characteristics including the response speed and amplitude of the piezo finger cannot be affected by the nonexistent heating problem. Thus, the operated samples of the PRH are also not affected by the heating problem.”

Comment 4. What is the duty of the finger? How many cycles can it work?

Response: We have fully considered this comment and replied to it from two aspects: one is a brief explanation about the duty of the finger, another is a brief discussion about its lifetime.

- (1) As for the duty of the piezo finger, it is designed to transform micro deformations of piezoelectric ceramic into multi-dimensional micro motions by converting electrical energy into mechanical energy. In detail, the compact configuration of integrating functional piezoelectric ceramic enables

the finger to produce 2-dimensional lateral bending motions and 1-dimensional longitudinal extending motion. These micro motions are used to plan functionalized hand gestures for further manipulating various objects. On the one hand, the piezo fingers are used to carry objects for manipulating them to produce multi-DOF motions by virtue of high structural stiffness. On the other hand, the piezo fingers can be used to carry grasping tools (for example, grasping needles in the supplementary experiments) to grasp some objects for transfer or other operations.

- (2) As for the lifetime of piezo finger, it should be noted that the lifetime of the piezo finger mainly depends on the lifetime of the crucial piezoelectric ceramic components due to its important roles of energy conversion and motion generation. Therefore, we analyze the lifetime from the piezoelectric ceramic. The piezoelectric ceramic used to construct the piezo fingers is PZT (Lead Zirconate Titanate ($\text{Pb}[\text{Zr}(x)\text{Ti}(1-x)]\text{O}_3$)) in this work. It is one of the most widely used piezoelectric ceramic materials. Some researches about the lifetime of PZT and the piezoelectric devices based on it have been implemented by some representative commercial companies and authoritative research institutions. Corresponding research results show that the lifetime of piezoelectric devices (for example, PZTs multilayer piezo stacks, Physik Instrumente, Germany) based on PZTs can perform 10 billion and even 100 billion cycles without loss of performance. NASA's research tests also validate that the performance of their actuator constructed on PZTs is over 100 billion cycles (*S. Sherrit et al., "Piezoelectric Multilayer Actuator Life Test," IEEE T Ultrason Ferr, vol. 58, no. 4, pp. 820-828, Apr 2011, doi: 10.1109/Tuffc.2011.1874.*). These results fully demonstrate that PZT is a kind of piezoelectric ceramic materials with long lifetime. Furthermore, we can also simply analyze the lifetime of the piezoelectric rings based on PZT from its working condition. The maximum excitation voltage of piezoelectric rings in this work is $600 V_{p-p}$, in which the thickness of single piezoelectric ring is 1 mm. This means that the maximum electric field strength is about 6×10^5 V/m. According to authoritative results that have been publicly reported, as for commercial piezo stack actuators with lifetime of 10 billion and even 100 billion cycles, their normal thickness of single layer of PZT is about $55 \mu\text{m}$, and their normal exciting voltage is about 150 V; this means that the corresponding electric field strength reaches 2.73×10^6 V/m. It can be found that the excitation strength of commercial piezoelectric products based on PZTs is significantly higher (about 4.55 times) than that of piezoelectric rings integrated in the piezo finger in this work. This means that the working condition of our piezo fingers is not strict as other piezoelectric devices with quite long lifetime.

In summary, above analyses and comparisons fully show that the key piezoelectric rings integrated in the piezo finger have lifetime in the same level with that of piezoelectric devices based on PZTs. That is to say the lifetime of the piezo finger can be estimated to more than 10 billion cycles conservatively. It is equivalent to continually operate about 10288 hours at the maximum operating frequency of 270 Hz and working voltage of $600 V_{p-p}$. In fact, we have accumulatively used our piezo robotic hand to carry out experimental research for at least 100 hours, in which no performance degradation has been found. In our follow-up research, we will consider continuing to study the life and reliability of our piezo robotic hand by a large number of experiments.

Comment 5. Will different amounts of piezo rings cause different response speeds and degrees? If so, give the plot.

Response: From two aspects: one is the influence of the number of the piezo rings on the response degree (i.e., output displacement at the fingertip) and another is the influence of the number of piezo rings on the response speed.

(1) The influence of the number of the piezo rings on the displacement at fingertip (by simulation):

According to your suggestions, we have analyzed the influence of the number of the piezo rings on the output displacement of the piezo finger by finite-element-method simulation with ANSYS. Meanwhile, the influence of the exciting voltage on the output displacement under different number of piezo rings has also been simulated. Take the lateral bending motion as an example, as plotted in the following Fig. R2-3, the output displacement of the piezo finger is approximately linear with the number of the piezo rings, which means that the output displacement can be adjusted to meet some potential demands by changing the number of piezo rings. It should be noted that the simulation results in two bending motion directions are exactly same due to structural symmetry. Quantitatively, the simulated lateral output displacement of the piezo finger is about $24\ \mu\text{m}$ under maximum exciting voltage of $600\ \text{V}_{\text{p-p}}$ when the number of the piezo rings are set as 20 PCS. When the number of the piezo rings increases to 2 times the situation used in this work (40 PCS), the output displacement ascends to about $58.6\ \mu\text{m}$ (relative to 2.44 times that under piezo rings of 20 PCS). Although the use of more piezo rings benefits to improving the output displacement, the capacitance will increase, which increases the power requirement of electrical exciting devices. Thus, the number of the piezo rings is just set as 20 PCS in this work to develop the prototype.

Fig. R2-3 The influences of the number of piezo rings and the exciting voltage on the output displacement of the piezo finger.

(2) The influence of the number of piezo rings on the response speed (by theoretical analysis):

The influence of the number of piezo rings on the response speed can be analyzed qualitatively from the electrical-mechanical conversion processes. Theoretically, piezo rings are typical capacitance components, and they can produce response by two processes: (i) the exciting voltage from the external excitation devices is applied to the piezo rings to charge them; (ii) the piezo rings produce conversion of electrical energy to mechanical energy for producing deformations. The piezo rings theoretically need charging time t_1 and conversion time t_2 to complete these two processes, respectively. It should be noted that the charging time t_1 is more dominant than time t_2 due to the fast conversion characteristic of the piezoelectric ceramic, which means that the response time of the piezo rings mainly depends on the charge time t_1 . As for the charge time t_1 of piezo rings, it mainly depends on the capacitance level of the piezo rings (determining how much energy needs to be charged) and the output current of the external excitation device (determining how fast to charge the energy). Single piezo ring is actually a parallel plate capacitance structure, and there is an electrical parallel relationship in multiple piezo rings. Therefore, the total capacitance will change when the number of the piezo ring changes, which affects the charge time t_1 and further affects the response speed of the piezo finger.

The capacitance level of the piezo rings can be described with intuitive mathematical expression as follows:

$$C_p = \sum_{i=1}^n \varepsilon \frac{S_i}{d_i} = n\varepsilon \frac{S}{d}$$

where, C_p is the total capacitance of one group of piezo rings used to generate 1-dimensional motion; n is the number of the piezo rings; S shows the relative electrode area of each piezo ring; d is the thickness of each piezo ring; ε is permittivity of piezo ceramic; it should be noted that the piezo rings are assumed to hold the same size (area S_i and thickness d_i) in the above equation.

The charging time of the piezo rings can be approximately expressed as follows:

$$t_1 \approx \frac{C_p U}{I} = n\varepsilon \frac{SU}{dI}$$

where, U and I are the exciting voltage applied on the piezo rings and output current of the external excitation device. The above equations indicate that the charging time t_1 is proportion to the number of piezo rings in theory. The total response time t can be represented as:

$$t = t_1 + t_2 \approx t_1 = n\varepsilon \frac{SU}{dI}$$

The conversion time t_2 can be ignored as it is short enough for piezoelectric ceramic, then the response speed can be regarded as only relation to charging time t_1 . In another word, under the same external excitation conditions, changing the number of piezo rings n will cause an approximately n times change in response time.

In general, the average response speed can be regarded as the ratio between the output displacement and the response time. It should be noted that the increase of the number of the piezo rings n not only leads to the increase of the output displacement, but also causes the increase of the response time. Therefore, the influence of increasing the number of the piezo rings n on the response speed can be analyzed by estimating the increase level of the output displacement and the response time. (i) According to the above equation about response time t , if the number of the piezo rings n increase to $2n$, the response time will increase to 2 times of that corresponding to the number of n . (ii) According to the simulation results shown in above Fig. R2-3, when the number of the piezo rings $n=20$ increases to $2n=40$, the output displacement will increase to about 2.44 times of that corresponding to $n=20$. These results show that the change level of the output displacement is more than that of the response time when changing the number of piezo rings n , which means that the increase of the number of piezo rings will lead to the increase of the response speed in theory (the conversion time t_2 is enough short to be ignored). Although the output displacement and the response speed can also be improved by increasing the number of the piezo rings, the resulting large capacitance will also increase the demand of output power to the external excitation system, especially for dynamic applications with high frequency. Thus, the number of piezo rings is a parameter that needs to be compromised in the development of the piezo robotic hand.

We believe that the above simulation and qualitative analyses results can illustrate the influence of changing the number of the piezo rings on the response degree (output displacement) and the response speed of the piezo finger.

Modification: According to your suggestion, the above Fig. R2-3 has been added into Fig. 2g of the revised manuscript, and a brief statement about the influence of changing the number of the piezo rings on the response displacement and response speed has been added into the supplementary information as follows.

“Supplementary Note 4

The influence of changing the number of the piezo rings on the response displacement and response speed of the piezo finger.

We also analyzed the influence of the number of the piezo rings on the output displacement of the piezo finger by finite-element-method simulation with ANSYS. Meanwhile, the influence of the exciting voltage on the output displacement with different number of piezo rings is also simulated. Take the lateral bending motion as an example, the output displacement of the piezo finger is approximately linear with the number of the piezo rings (Fig. 2g), which means that the output displacement can be adjusted to meet some potential demands by changing the number of piezo rings. It should be noted that the simulation results are exactly the same in two bending motion directions due to structural symmetry. Quantitatively, the simulated lateral output displacement of the piezo finger is about 24 μm under the maximum exciting voltage of 600 V_{p-p} when the number of the piezo rings are set as 20 PCS. When the number of the piezo rings increases to 2 times the situation used in this work (i.e., 40 PCS), the output displacement ascends to about 58.6 μm (relative to 2.44 times that under piezo rings of 20 PCS).

The influence of the number of piezo rings on the response speed can be analyzed qualitatively from the electrical-mechanical conversion processes. Theoretically, piezo rings are typical capacitance components, and they can produce response by two processes: (i) the exciting voltage from the external excitation devices is applied to the piezo rings to charge them; (ii) the piezo rings produce conversion of electrical energy to mechanical energy for producing deformations. The piezo rings theoretically need charging time t_1 and conversion time t_2 to complete these two processes, respectively. It should be noted that the charging time t_1 is more dominant than time t_2 due to the fast conversion characteristic of the piezoelectric ceramic, which means that the response time of the piezo rings mainly depends on the charge time t_1 . As for the charge time t_1 of piezo rings, it mainly depends on the capacitance level of the piezo rings (determining how much energy needs to be charged) and the output current of the external excitation device (determining how fast to charge the energy). Single piezo ring is actually a parallel plate capacitance structure, and there is an electrical parallel relationship in multiple piezo rings. Therefore, the total capacitance will change when the number of the piezo ring changes, which affects the charge time t_1 and further affects the response speed of the piezo finger.

The capacitance level of the piezo rings can be described with mathematical expression as follows:

$$C_p = \sum_{i=1}^n \varepsilon \frac{S_i}{d_i} = n\varepsilon \frac{S}{d} \quad (3)$$

where, C_p is the total capacitance of one group of piezo rings used to generate 1-dimensionanl motion; n is the number of the piezo rings; S shows the relative electrode area of each piezo ring; d is the thickness of each piezo ring; ε is permittivity of piezo ceramic; it should be noted that the piezo rings are assumed to hold the same size (area S_i and thickness d_i) in the above equation.

The charging time of the piezo rings can be approximately expressed as follows:

$$t_1 \approx \frac{C_p U}{I} = n\varepsilon \frac{SU}{dI} \quad (4)$$

where, U and I are the exciting voltage applied on the piezo rings and output current of the external excitation device. The above equations indicate that the charging time t_1 is proportion to the number of piezo rings in theory. The total response time t can be represented as:

$$t = t_1 + t_2 \approx t_1 = n\varepsilon \frac{SU}{dI} \quad (5)$$

The conversion time t_2 can be ignored as it is short enough for piezoelectric ceramic, then the response speed can be regarded as only relation to charging time t_1 . In another word, under the same external excitation conditions, changing the number of piezo rings with n times will cause an approximately n times change in response time.

In general, the average response speed can be regarded as the ratio between the output displacement and the response time. It should be noted that the increase of the number of the piezo rings n not only leads to the increase of the output displacement, but also causes the increase of the response time. Therefore, the influence of increasing the number of the piezo rings n on the response speed can be analyzed by estimating the increase level of the output displacement and the response time. (i) According to the above equation about response time t , if the number of the piezo rings n increase to $2n$, the response time will increase to 2 times that corresponding to the number of n . (ii) According to the simulation results shown in Fig. 2g, when the number of the piezo rings $n=20$ increases to $2n=40$, the output displacement will increase to about 2.44 times of that corresponding to $n=20$. These results show that the change level of the output displacement is more than that of the response time when changing the number of piezo rings, which means that the increase of the number of piezo rings leads to the increase of the response speed in theory.”

Comment 6. “With the special structural and electrical configurations of the actuation part, it can produce bidirectional bending motions and extending motions.” What is the different electrical configuration? At least show the entire details of one deformation type.

Response: The purpose of this statement is to illustrate that the piezo finger can produce bending and extending motions by using the configurations with a group of piezo rings, in which each piezo ring holds four polarized regions and can be excited by applying different signals to each polarized region. The configurations and multi-dimensional deformations of the actuation part integrating piezo rings with four polarized regions are shown in the following Fig. R2-4. The special things mean that the four-region polarization and combined excitation scheme are used to generate bending and extending motions. The detailed descriptions about this configuration have been provided in the supplementary information.

Fig. R2-4 Electrical configurations and multi-dimensional deformations of the actuation part integrating piezo rings with four polarized regions. (a) Polarization configurations of adjacent two piezo rings with four polarized regions. (b) Structural configurations of adjacent two piezo rings. (c) Bending and extending deformations of the whole actuation part.

In fact, we can also use other polarization configurations and schemes to realize similar motions. A representative configuration is shown in the following Fig. R2-5. In detail, as shown in Fig. R2-5a, two groups of piezo rings (named as top half and bottom half) can be configured to form an actuation unit, in which the top half and the bottom half contain several actuation unit-I and several actuation unit-II, respectively. The copper electrode slices are set between the adjacent two piezo rings to apply electric fields on them. It should be noted that each of piezo rings in unit-I and piezo rings in unit-II holds two polarized regions, and the piezo rings in unit-I and that in unit-II are configured to mutually orthogonal. The more details are shown in Fig. R2-5b. With these configurations, the top half and the bottom half can separately produce 1-dimensional bending deformation when the same electric fields are applied on the two groups of opposite regions simultaneously, while they can also produce 1-dimensional extending

deformation when the opposite electric fields are applied on the two groups of opposite regions simultaneously. Therefore, the whole actuation part integrating piezo rings with two polarized regions can produce bending deformations along x and y axes, and extending deformation along z axis, as shown in Fig. R2-5c. It can be found that this configuration will lead to response difference in the two bending deformations. This is because that there is an amplification effect (by the height of the top half of the actuation part) on the response when the bottom half of the actuation part is stimulated (assuming to excite bending deformation along x axis), whereas there is no amplification effect when the top half is stimulated (assuming to excite bending deformation along y axis). Therefore, in order to avoid the response difference in the two lateral bending motions of a piezo finger when using the same amount of piezo rings to stimulate motions, the configurations of using piezo rings with four polarized regions are utilized in this work.

Fig. R2-5 Configurations and multi-dimensional deformations of the actuation part integrating piezo rings with two polarized regions. (a) Configurations and electrode definitions. (b) Configurations of adjacent two piezo rings with two polarized regions. (c) Bending and extending deformations of the whole actuation part.

Modification: According to your suggestion, the entire details of the configurations shown in above Fig. R2-5 have been added into the Supplementary Fig. 2h to Fig. 2j, and a corresponding description has been added into the Supplementary Note 2 of the revised manuscript as follows.

“...In fact, we can also use other polarization configurations and schemes to realize similar motions. A representative configuration is shown in the Supplementary Fig. 2h to Fig. 2j. In detail, as shown in Supplementary Fig. 2h, two groups of piezo rings can be configured to form an actuation unit (named as top half and bottom half), in which the top half and the bottom half contain several actuation unit-I and several actuation unit-II, respectively. The copper electrode slices are set between the adjacent two piezo rings to apply electric fields on them. It should be noted that each of piezo rings in unit-I and piezo rings in unit-II holds two polarized regions, and the piezo rings in unit-I and that in unit-II are configured to mutually orthogonal. The more details are shown in Supplementary Fig. 2i. With these configurations, the top half and the bottom half can separately produce 1-dimensional bending deformation when the same electric fields are applied on the two groups of opposite regions simultaneously, while they can also produce 1-dimentional extending deformation when the opposite electric fields are applied on the two groups of opposite regions simultaneously. Therefore, the whole actuation part integrating piezo rings with two polarized regions can produce bending deformations along x and y axes, and extending deformation along z axis, as shown in the Supplementary Fig. 2j. It can be found that this configuration will lead to response difference in the two bending deformations. This is because that there is an amplification effect (by the height of the top half of the actuation part) on the response when the bottom

half of the actuation part is stimulated (assuming to excite bending deformation along x axis), whereas there is no amplification effect when the top half is stimulated (assuming to excite bending deformation along y axis). Therefore, in order to avoid the response difference in the two lateral bending motions of a piezo finger when using the same number of piezo rings to stimulate motions, the configurations of using piezo rings with four polarized regions are utilized in this work.”

Comment 7. “The tested results show that the velocity ascends with the increase of frequency within specific ranges, while it descends after the frequency increases to some key frequencies.” Why? What is the key frequency?

Response: This statement refers to the experimental results of the velocity versus the exciting frequency, which means that the manipulation velocity changes from ascending trend to descending trend when the exciting frequency increases to over than a specific frequency. The exciting frequency corresponding to obtain peak velocity means the key frequency because that the velocity trend produces change when the exciting frequency exceeds this value.

The reason for causing velocity decrease when frequency increase can be attributed to the dynamic and static alternate friction working mechanism. The motion displacement during one period of the dynamic and static friction manipulation process is shown in the following Fig. R2-6. S_1 and S_2 represent displacements of the static friction and dynamic manipulation stages, respectively. With the increase of the exciting frequency, the time occupied by the static friction stage in one exciting period becomes more and more short, and the sliding motion between the fingertips and the manipulated object will occur when the time of this stage is shortened to a certain extent. This sliding motion results in the decrease of the displacement of static friction stage S_1 , which causes the decrease of the effective displacement step pitch ΔS in single manipulation period, and the manipulation velocity is decreased resultantly. In fact, the relative sliding motion in the static friction action stage will be aggravated gradually when the exciting frequency further increases. Thus, the manipulation velocity decreases when the exciting frequency increases to more than the “key frequency”.

Fig. R2-6 Diagram for the exciting signal and displacement of the static and dynamic manipulation processes

Modification: In order to improve the rigor of this article, a brief statement to explain the phenomenon of velocity decrease at high working frequency has been added into the revised manuscript as follows:

“Notably, with the increase of the exciting frequency, the time occupied by the static friction stage in one exciting period becomes more and more short, and the sliding motion between the fingertips and the manipulated object will occur when the time of this stage is shortened to a certain extent. This sliding motion results in the decrease of the displacement of static friction action stage, which causes the decrease of the effective displacement step pitch in single manipulation period, and the manipulation velocity decreases resultantly.”

Comment 8. “The no-load velocities of LX_p, LY_p, and RZ_p DOFs are 18.73 μm/s, 19.74 μm/s” why is the velocity different since the ring is symmetrical?

Response: As illustrated in the response to above comment 7), the manipulation velocity depends on the effective displacement step pitch ($\Delta S=S_1-S_2$) under specific exciting frequency. In principle, the manipulation displacement of the static friction stage S_1 relies on the output displacement of the piezo fingers at the fingertips. It should be noted that the LX_p and LY_p DOFs are manipulated with different hand gestures, in which LX_p uses the gesture that all piezo fingers bend along X axis and LY_p uses the gesture that all piezo fingers bend along Y axis. According to the manipulation mechanisms and relationships shown in the above Fig. R2-6, the manipulation velocity results from multiplying the effective motion step pitch and the manipulation frequency. The effective motion step pitch ΔS relies on the forward displacement S_1 in the static friction stage and the backward displacement S_2 in the dynamic friction stage. Therefore, the velocity difference can be analyzed from the following two aspects.

(1) One aspect is from the response characteristics of the piezo fingers in X axis and Y axis. Although the structural and electrical configurations of piezo finger are symmetrical in the orthogonal X axis and Y axis, the response difference between these two directions are unavoidable due to assembly errors, especially for assembly process of multiple piezo rings. Furthermore, the core piezo rings with four polarized regions are used to transform electrical energy into mechanical energy. It should be noted that a necessary step during processing of piezoelectric ceramics is to conduct polarization treatment under long-time and high-strength polarization electric field, which is used to ensure that the piezoelectric ceramics have sufficient strength of piezoelectric effect. This process may lead to different piezoelectric characteristics at different polarized regions. Therefore, when the same excitation conditions are applied on different polarized regions, their responses may have some differences. In short, both the machining and assembly errors may lead to the differences of the piezo finger in X and Y directions, and they further cause tiny difference on manipulation velocities.

(2) Another aspect is from the motion transformation from the piezo fingers to the manipulated object. The backward displacement S_2 in the dynamic friction stage depends on the relative sliding motion between the piezo fingers and the manipulated object. This means that the difference of friction behavior (related to friction coefficient and contact state) between the piezo fingers and the manipulated object in X and Y axes may lead to difference of backward displacement S_2 in the dynamic friction stage for the two motion directions, which affects the effective motion step pitch ΔS and further influences the manipulation velocity.

In summary, both the fabrication errors of the piezo fingers and the friction characteristics between the piezo fingers and the manipulated object in different directions are the factors to cause the difference of manipulation velocities in diverse directions. Based on the above analyses, the minor differences of velocity characteristics are unavoidable in practice, but these differences are with a reasonable and acceptable range.

Modification: A brief statement about the causes of velocity difference in different DOFs has been added into the revised manuscript as follows:

“The tiny difference between the manipulation velocities in X axis and Y axis can be mainly attributed to two factors. One is the different response characteristics of piezo finger in X and Y axis, which is mainly caused by machining and assembly errors of the piezoelectric rings. Another is that the frictional characteristics between the piezo fingers and the manipulated object are not exactly same in X axis and Y axis.”

Comment 9. The finger design and working mechanism have been reported in the publication, Bioinspired Multilegged Piezoelectric Robot: The Design Philosophy Aiming at High-Performance Micromanipulation. Please describe in detail the novelty of this manuscript.

Response: Your concern on the novelty of this article is important, and we agree that different research focuses between this work and our previous work should be clarified in detail. We briefly explain the differences and connections between this work and our previous work from aspects of focus, design, mechanism, function and application.

(1) From the focus aspect:

In the previous work, our focus is to develop a bioinspired hexapod piezoelectric robot and investigate its performance integration of high precision, long stroke, strong carrying capability, and multi-DOF motions. It is worth noting that the motion characteristics on a specific working plane of the robot itself are investigated in detail. The main focus is the motion ability of the robot itself.

In this work, our focus is to explore new design of robotic hand with different principles, configurations and methods for solving the problems of low manipulation accuracy, poor electromagnetic compatibility and complex system of other robotic hands due to their limitations in structures, principles and transmissions. The further purpose is to investigate the motion manipulation characteristics, manipulation capabilities and effects for various objects with different shapes, materials and dimensions. Our main focuses are not only on the fundamental characteristics of the piezo robotic hand, but also on the manipulation capability (including grasping operation) to other various objects. Furthermore, lots of application explorations are focused to demonstrate manipulation capability and practical value of the piezo robotic hand. In contrast, the research focuses of this work are different from that of our previous work.

(2) From the design and mechanism aspects:

As for the hexapod piezoelectric robot developed in the previous work, it holds six piezoelectric robot leg (PER-leg). Each PER-leg utilizes a bending stack and a stretching stack to generate bending motions and extension motion, respectively. The use of the stretching stack leads to high capacitance of the PER-leg, which brings higher requirement to the electrical excitation devices in output power and increases the costs in PER-leg itself and its use cost. Besides, it should be noted that the ring-shape piezoelectric ceramics are just used to produce bending motions, which does not fulfil its full potential function. Furthermore, the use of stretching stack needs more design on preload mechanism for applying preload force to ensure normal working condition, and the lateral force acting on the stretching stack may bring damage. Therefore, in this work, the lateral bending and longitudinal extending motions of the piezo fingers are realized by planning different excitation combinations of the piezoelectric ceramic rings. Using the same set of piezoelectric ceramic rings to achieve bending and extending motions, the structures of the piezo fingers are ensured to be as simplified as possible, and the cost of the piezo fingers can be reduced as far as possible. In short, our design of piezo finger in this work carefully considers the control of structural complexity, manufacturing and use costs.

As for mechanism, our previous work focuses on wobbling, slipping and walking gaits of the hexapod piezoelectric robot to realize high resolution and large range movement of the robot itself. In contrast, this work focuses on motion manipulation of various other objects (not the piezo robotic hand itself) with different shapes, materials and dimensions by alternative dynamic and static friction actions and diverse functionalized gestures. The common connection of the mechanisms is just based on frictional actions. On the other hand, this work also focuses on the direct action of specific

functionalized gesture for grasping operation of some small objects (as illustrated in the supplementary grasping experiment).

(3) From the function aspect:

In the previous work, the developed hexapod piezoelectric robot can only realize its own motion on a specific working plane. It can only be used to carry objects to move or rotate on the plane, and the carried objects can be operated with external manipulation tools or devices. The motion ranges and forms of the hexapod piezoelectric robot are limited by the working plane. Note: the developed hexapod piezoelectric robot moves by itself, not to manipulate other objects.

In this work, the developed piezo robotic hand is not limited by the working configuration (for example, on a specific plane). It can achieve abundant functionalized gestures by separately exciting its four piezo fingers under any working configuration, for example, working as a stand-alone device or installing on the end of a 6-DOF mechanical arm as the execution end. With the preplanned functionalized gestures, our piezo robotic hand can not only manipulate various objects to achieve multi-DOF and multi-form motions, but also grasp some objects (for example, small sphere with diameter of 1.2 mm and a gravel with irregular shape). In short, the developed piezo robotic hand is used to directly manipulate or grasp other objects, which means that the function targets of this work are different from that of our previous work.

(4) From the application aspect:

In the previous work, we just demonstrate the precision positioning of the developed hexapod piezoelectric robot for batch injection of multiple cells and micromachining on large surfaces. In this work, we not only study the manipulation capability of the developed piezo robotic hand to various objects, but also further study the feasibility to construct specific application system with diverse functions. Furthermore, the grasping application is also demonstrated. In short, the study contents in application aspects of this work are different from that of our previous work.

In summary, we believe that this work and our previous work are independent in main contents. According to your constructive suggestions and the professional comment 1 of Reviewer #1, the main novelties and contributions of this work are concluded with bullets in the revised manuscript, which we hope meet with your approval and give the reader a clear explanation.

Modification: According to the professional suggestions of you and Reviewer #1, the novelties of this work are summarized with bullets (i), (ii), (iii) and (iv) in the Introduction of the revised manuscript as follows:

“The main novelties and contributions of this work can be concluded as: (i) This work proposes the first robotic hand with four fingers constructed on piezoelectric ceramics, it achieves in-hand multi-DOF manipulation, as well as high resolution of 15 nm, fast response of 0.5 ms, low hysteresis of 3.95% and no electromagnetic interference. (ii) The PRH is designed to be a unique rigid configuration with four bolt-clamped metal-ceramic sandwich piezo fingers, which achieves compact structures and large load ability as it has no transmission mechanisms and hinges. (iii) The PRH achieves motion expansion from the micro deformations of piezoelectric ceramics to the macro motions of various objects by four-finger cooperative manipulations, and realizes multi-DOF and cross-scale motion manipulation. (iv) The PRH realizes excellent adaptability to manipulate various objects with diverse shapes, materials, and dimensions. A series of experiments demonstrate its great application potentials to construct multi-DOF manipulation devices and perform grasping operations.”

Comment 10. In fig. 2, a scale bar may be helpful to show the size of the hand.

Response: According to your professional suggestions, a scale bar with length of 10 mm has been added into Fig. 2a in the revised manuscript.

Revised Fig. 2 in the revised manuscript (the scale bar has been added into a).

Responses to reviewers' comments (Reviewer: #3)

Comments to the Author:

This paper presents the development and utility of a linearly moving, highly rigid piezo-robot hand (PRH) made of functional piezoelectric ceramics.

The paper shows that the four piezoelectric fingers enable 12-DOF motion, which is difficult to achieve with existing actuators. Notably, they also show in a movie that 18 motions and 3R motions are generated by the motion of a sphere, including plate manipulation that realizes 2L (linear) and 1R (rotational) motions and cylindrical object manipulation that realizes 1L and 1R motions, and report the world's first multi-degree-of-freedom ultra-precision manipulation device.

Actuators are positioned as extremely important devices along with sensors in the next generation digital society. However, compared to sensors, research and development of actuators is more difficult and progress has been slow. In particular, with actuators, there is a trade-off between response speed, force generated, amount of displacement generated, and drive voltage. Therefore, electrical operation has limited applications, requiring complex mechanical structures such as pneumatic, hydraulic, or cable actuation.

The authors' proposal is electrically operable and resistant to electromagnetic noise, enabling extremely delicate operation, and has an extremely high generating force, which is a characteristic of piezoelectric ceramics.

The video demonstration shows the operation of very large and heavy objects as well as lightweight and cylindrical objects, and the range of applications is wide. The detailed analysis of the mechanisms and numerical analysis related to the operation is also provided, which is of great value not only in the applied aspect but also in the academic aspect.

I believe that publication in Nature Communications is appropriate after answering the following minor.

Response: Thank you very much for taking the time to review our manuscript. Your positive and instructive comments encourage our work to a great extent. Your opinions are not only helpful for us to improve this article, but also of great guidance to our future work. The manuscript has been carefully revised according to your suggestions. All of the modifications have been highlighted with yellow tool in the revised manuscript and supplementary information. We have carefully replied to your comments point-to-point and as detailed as possible in the following. Meanwhile, the main modifications are attached below in yellow highlight for your fast tracking.

Comment 1. The experimental results and videos are excellent and show that the behavior described in the paper is faithfully reproduced. On the other hand, there are many illustrations, etc., and almost no photos of actual device structures and components. Actual photographs should be shown along with the illustrations. In particular, it would be academically preferable for Figures 2 and 3 to be shown together with photographs rather than illustrations.

Response: Your suggestions are very instructive for us to improve the reasonability and formalization of the Figures. We have carefully considered Figures 2 and 3 to add corresponding photographs of entire prototype and components. It should be noted that the photos of the whole PRH prototype and its components can be easily obtained by general photographic devices. Therefore, the photographs of the actual PRH prototype (packaged and unpackaged states), actuation part and piezo fingers have been added into Fig. 2 according to your professional suggestions. However, the photos of the functionalized gestures shown in Fig. 3 are difficult to obtain due to micro deformations. In detail, the deformations or motions of the piezo fingers are within micro scale, which cannot be entirely captured by current photographic devices. (Current microscopic imaging devices cannot capture entire gestures of piezo fingers with centimeter size level). This is why we use some exaggerated diagrams to show gestures in Fig. 3 and Supplementary Fig. 5.

Therefore, Fig. 3, we have kept using the illustrations to clearly show the micro motions of piezo fingers, in which the micro photos of the four fingertips have been added into Fig. 3h. Furthermore, a functionalized gesture for grasping operations has been added into Fig. 3g.

As for other Figures, we have also carefully considered to add photographs of the actual devices. For example, the photographs of the actuation unit and components (piezoelectric ceramic ring, common electrode and fan-shaped electrode) have been added into Supplementary Fig. 2g. The photographs of experimental devices have been also added into Supplementary Fig. 8 (Supplementary Fig. 8c for testing noise characteristic, Supplementary Fig. 8d for testing thermal characteristic, Supplementary Fig. 8f and Fig. 8g for grasping experiments).

Modification: The revised Fig. 2, Fig. 3, Supplementary Fig. 2 and Supplementary Fig. 8 are as follows. The related citations have been also updated in the revised manuscript and supplementary information.

Revised Fig. 2 in the revised manuscript: the photos of the PRH prototype, the actuation part and the piezo fingers are shown in a and d.

Revised Fig. 3 in the revised manuscript: a functionalized gesture for grasping operations has been added into g and the micro images of the four fingertips have been added into h.

Revised Supplementary Fig. 2 with the revised manuscript: the photographs of the actuation unit and components have been added into (g); h, i and j have been added for illustrating other configurations of the actuation part.

Revised Supplementary Fig. 8 with the revised manuscript: some photos of the experimental devices have been added into c, d, e, f and g.

Comment 2. I believe that the object to be compared should be a precision stage, not a robotic hand. In fact, researchers studying robotic hands are aiming at flexible mechanisms, force generation, displacement control, etc. for softly grasping eggs and other objects without breaking them, and not at the movements shown in the movies in this paper and others. The performance comparisons shown in Figure 6 and elsewhere are not accurate. If the proposed actuator is to be compared to a robotic hand, a demonstration video should be shown for comparison, such as "grabbing things".

Response: This comment is of great instructive significance, and we fully agree with the research conventions about robotic hands. In this work, the research focus of piezo robotic hand is actually with some differences to the traditional robotic hands. We reply to this comment in detail with the following several aspects, including background, motivation, grasping experiments (suggested by you and Reviewer #1) and our consideration about the compared objects.

(1) Background

The purposes of many researchers to develop various robotic hands are usually to replace or assist human hand for carrying out general repetitive tasks, and the related researches are aiming at several focuses that you pointed out. One of the most typical applications of robotic hands is to grasp something and move it from one position to another one (this can be called "off-hand manipulation"). It is also the

most common jobs of human hands in our daily life. However, it should be noted that human hand can not only execute various grasping operations, but also hold good ability to implement flexible attitude adjustment of an object in the hand (for example, we pick up some components of a watch to carry out repair and assembly.). We can call this application “in-hand manipulation”. For this kind of application requiring high manipulation precision and flexible manipulation DOF, the existing various robotic hands are difficult to be competent due to their limitations in structures and principles.

(2) Motivation

Based on the inverse piezoelectric effect of piezoelectric ceramics, we designed a novel rigid robotic hand with four piezo fingers to achieve adaptability, multi-DOF and high resolution object manipulations in this work. The core idea is to use multi-dimensional and high-resolution micro deformations of a group of piezoelectric rings to make the piezo finger have flexible motions (2D bending motions and 1D extending motion) through structural and electrical designs. This work fully demonstrates the idea of motion manipulation from micro to macro by using piezoelectric ceramics. It holds difference by comparing with traditional robotic hands in terms of structural features, working principles and functional characteristics. Based on the above analyses, we cautiously believe that this work provides a new exploration case to develop robotic hand by using smart functional material and new working principle. Piezoelectric drive is just the actuation type of our robotic hand (used to replace traditional actuation types liking electromagnetic, fluid, etc.), and our purpose is to achieve “in-hand manipulation” instead of human hand or even over human hand.

In fact, the developed piezo robotic hand not only holds excellent capability of completing in-hand motion manipulations, but also can achieve grasping operations, especially to grasp micro-objects. According the professional suggestions of you and Reviewer #1, the grasping experiments of our piezo robotic hand has been carried out and added into the revised manuscript.

(3) Supplementary grasping experiments according to suggestions of you and Reviewer #1.

As mentioned in the previous manuscript, more functionalized gestures can be obtained by using motion combinations of the four piezo fingers. Thus, a functionalized gesture for grasping operations has been planned and added into the revised manuscript, as shown in the revised Fig. 3g. The packaged prototype of the piezo robotic hand can be installed on the end of a 6-DOF mechanical arm to achieve function extension. For example, extending manipulation DOF of the mechanical arm and manipulating object for in-situ motion, as the experimental records shown in Supplementary Movie 9 and revised Fig. 5h. In order to further demonstrate the grasping operation of the piezo robotic hand, as shown in revised Fig.5i, it is installed on the end of the mechanical arm, and four levers are installed on the four piezo fingers to extend the motion range of the fingertips. Then, four grasping needles are fixed on the ends of the four levers, in which the tips of the four grasping needles are adjusted to close with each other. Several small sphere (with diameter of 1.2 mm) and a pile of gravels (size within 1.5 mm) are placed on a platform, then the piezo robotic hand is used to grasp a sphere and a gravel from some of them, as shown in revised Fig. 5j. The complete experimental setup is illustrated in the revised Supplementary Fig. 8f and Fig. 8g. The grasping needles can be adjusted to close the target sphere or gravel by the control panel, then the target sphere is successfully grasped from one position and moved to another position by releasing with the piezo robotic hand. Similarly, a gravel is also grasped from a pile of gravels. The video sequences for grasping a small sphere and a gravel are shown in revised Fig. 5k, and the corresponding experimental records are provided in Supplementary Movie 10. These grasping experiments fully demonstrate that the developed piezo robotic hand holds great application potentials for robotic grasping operations, which we hope meet with your approval. Some contents about the grasping experiments have been added into the manuscript. In the future, we will focus on deep researches in grasping structures, strategies, control methods, etc.

(4) Our considerations about the compared objects.

Based on the above analyses and experimental demonstrations, we intend to remain the statement of “piezo robotic hand”. We believe that the ideas and designs in this work are with referential significance to the development of robotic hand. On the one hand, we quantitatively compare the fundamental characteristics of the developed piezo robotic hand with other robotic hands, so as to highlight its features. On the other hand, considering that we use the piezo robotic hand to manipulate a plate for realizing multi-DOF motions, we also add a brief comparison between the plate manipulated with our piezo robotic hand and other precision stages according to your suggestions, as replied to your comment 3).

Modification: According to the professional suggestions of you and other two Reviewers, the grasping experiment of our robotic hand has been carefully planned and implemented to demonstrate its application potentials and the related descriptions have been added into the revised manuscript. The quantitative comparisons between the developed piezo robotic hand and other robotic hands have been added into the revised manuscript.

(1) About the grasping experiments

Revised Fig. 5 in the revised manuscript (See i, j and k to know the configurations and results of grasping experiments)

“...In order to further demonstrate the grasping operation of the PRH, it is installed on the end of a 6-DOF mechanical arm, and four levers are installed on the four piezo fingers to extend the motion range of the fingertips. Four grasping needles are fixed on the ends of the four levers, in which the tips of the four grasping needles are adjusted to close with each other, as shown in Fig. 5i. Several small spheres (with diameter of 1.2 mm) and a pile of gravels (size within 1.5 mm) are placed on a platform, and the PRH is used to grasp a sphere and a gravel from the placed targets, as shown in Fig. 5j. The complete experimental setup is illustrated in Supplementary Fig. 8f and Fig. 8g. The grasping needles can be adjusted to close the target sphere or gravel by the control panel, then the target sphere is successfully grasped from one position and moved to another position by releasing with the PRH. Similarly, a gravel is also grasped from a pile of gravels. The video sequences for grasping a small sphere and a gravel are shown in Fig. 5k, and the complete experimental records are provided in Supplementary Movie 10. These grasping experiments fully demonstrate that the developed PRH holds great application potentials for robotic grasping operations.”

(2) About the quantitative comparison between the fundamental characteristics of our PRH and other robotic hands:

“Fig. 6. Characteristic summary and comparison of the PRH. a Characteristics of the PRH summarized from three aspects: (i) structures and principles; (ii) fundamental characteristics; (iii) manipulation characteristics. b characteristic comparison of the PRH and other robotic hands. c Main advancements of this work compared with other robotic hands.”

“...The characteristics of our PRH can be concluded from aspects of structures and principles, fundamental and manipulation characteristics (see Fig. 6a). In order to evaluate the characteristic level of

our PRH, a quantitative comparison about the response time, resolution or accuracy, response hysteresis, and frequency feature of our PRH and other robotic hands is accomplished (Fig. 6b). It can be found that the response time of the PRH is obvious short than other robotic hands due to the fast response capability of the functional piezoelectric ceramics. Our PRH achieves excellent open-loop motion resolution of about 15 nm, which is difficult for other robotic hands. With the inherent fast electromechanical conversion of the piezoelectric ceramics, our PRH achieves low hysteresis of 3.95% that obviously lower than other robotic hands. Furthermore, owing to the unique design of high rigidity and no transmission structures, our PRH achieves high natural frequency of about 4.1 kHz; this provides a good bandwidth base for high working frequency, which is superior to other low-rigidity robotic hands. In short, our PRH holds merits including low hysteresis (<3.95%), high resolution (15 nm), fast response (0.5ms) and high natural frequency (4.1kHz).”

Comment 3. The piezoelectric device proposed in this paper is an excellent actuator, and the detailed description of the device structure is valuable both academically and in terms of application. On the other hand, I do not think it is appropriate in comparison with a robotic hand. If a comparison with a robotic hand is to be made, such a demonstration or the like should be shown, but at present it is only used as a "precision stage".

Response: Thanks a lot for your instruction and reminder for us to add relevant contents for supporting the core concept of “piezo robotic hand” in this article. As illustrated in our response to your comment 2), most researchers have developed robotic hands inspired by structural and functional inspirations of human hands. In fact, both the “off-hand manipulation” (grasping operation, slapping, etc.) and “in-hand manipulation” are the basic functions of human hands. The former means that various objects are usually grasped in one position and moved to another position. The latter means that a specific object can contact with hand for realizing high flexibility and high precision attitude manipulation. Lots of literature reports show most previous researches about robotic hands mainly aim at the “off-hand manipulation”, and the most typical case is to grasp some objects, including fruits, parts, and so on. At the same time, the “in-hand manipulation” of various objects to achieve multi-DOF and high precision motions is also a significant research topic, while other robotic hands usually pay less attention to “in-hand manipulation”, especially for high manipulation precision. Furthermore, other robotic hands are difficult to implement in-hand precision manipulation due to their limitations in structural configurations and working principles.

To sum up, it is of great significance to develop a robotic hand that can achieve in-hand precision manipulation, which is why we carried out this work. Most importantly, our extensive literature researches show that “robotic hand” is a relatively wide concept. In another word, lots of prototypes using the idea of structural and functional simulations are widely called as “robotic hand”, in which their shapes, dimensions, structures, functions are diversified. We believe that the piezo robotic hand in in this work adopts a completely different principle from the traditional robotic hands, in which complex transmission mechanisms are avoided completely. From the aspect of operation functions, our piezo robotic hand not only holds excellent capability of completing in-hand motion manipulations, but also can achieve grasping operation similar to other robotic hands.

As for the concept of precision stage, one fact should be noted that manipulating flat plate object to achieve multi-DOF motions is just one of the functions of our PRH. We just take it as an example to illustrate the manipulation capability of our PRH. It holds other functions of manipulating diverse objects and grasping micro objects. According to the professional suggestions of you and Reviewer #1 and considering the article length simultaneously, we have made three improvements: (1) the grasping application experiments of our piezo robotic hand has been carried out to demonstrate its operation

capability liking other robotic hands. (2) The quantitative comparison between the developed piezo robotic hand and other robotic hands has been added into the revised manuscript. (3) A brief characteristic comparison between the plate manipulated with our PRH and other precision stages have been implemented to add in the supplementary information. We hope these improvements can meet with your approval.

Modification:

(1) The grasping experiment of our piezo robotic hand has been carried out supplementarily, and the corresponding descriptions, fingers and movies have been added in to the revised manuscript and supplementary materials. More details can be found in the responses to your comment 2).

(2) The quantitative characteristic comparison between our PRH and some other robotic hands has been added into the revised manuscript, as illustrated in the responses to your comment 2).

(3) A brief comparison between the plate manipulated with our PRH and other precision stages has been added into the supplementary information as appended below.

“Supplementary Table 3

Characteristic comparison between the plate manipulated with the PRH and other precision stages

Item	Compliant platform¹	Micro-positioning stage²	Nano-positioning stage³	Nano-positioning stage⁴	Nano-positioning platform⁵	Micro/nano positioning stage⁶	Plate manipulated with the PRH
DOF	2R	3L	2L	2L	3L+3R	2L	2L+1R
Principle	Piezo	Piezo	Piezo	Piezo	Electromagnetic	Electromagnetic	Piezo
Overall size	30×41×41 mm ³	77×77×77 mm ³	NA	NA	250×250×57.4 mm ³	NA	φ85mm×124 mm
Motion stroke	2.04mrad ×2.12mrad	582μm ×517μm ×524μm	1.035mm×1.035mm	42μm×42μm	0.5mm×0.5mm×5 mrad	2.13mm×2.02 mm	25mm×25mm×2πrad
Working voltage	100V	100V	150V	150V	48V	NA	600V _{pp}
Working frequency	100Hz	69Hz	32Hz	100Hz	<50.8Hz	<43.7Hz	270Hz
Load capability	0.25kg	NA	0.1kg	NA	0.9kg	2kg	14.76kg

Notes: “NA” means the compared item is not available.

“Supplementary Note 9

Characteristic comparison between the plate manipulated with the PRH and other precision stages

Manipulating the plate to produce two translational DOFs and one rotary DOF motions is an important capability of our PRH, which is experimentally evaluated in detail in this work. In order to further evaluate the level of its manipulation characteristics, a simple characteristic comparison between the motion plate manipulated with our PRH and other precision stages is accomplished (Supplementary Table 3). The compared items contain the motion DOF, principle, overall size, motion stroke, working voltage, working frequency and load capability. The compared results show

that the motion plate manipulated with our PRH holds several merits: (i) the motion plate manipulated with our PRH achieves greater motion strokes; (ii) the working frequency is more than other motion stages; (iii) the load capability of the motion plate manipulated with our PRH achieves excellent capability to carry other objects for motions. It is worth noting that the plate manipulated with our PRH is only a construction case of many promising functions. In the follow-up work, we will also consider using the PRH to build multi-DOF devices for specific applications.”

REVIEWERS' COMMENTS

Reviewer #2 (Remarks to the Author):

Very nice work! I am happy with the revision; thus, I recommend publication to Nature Communications.

Reviewer #3 (Remarks to the Author):

The revised paper based on the reviewer's comments is excellent, supported by very substantial quantitative data and experimental results. In particular, it clearly demonstrates the usefulness and novelty of the newly developed robotic hand. The revised paper is worthy of publication because it provides important guidance to many researchers and is also useful to industry.

Responses to reviewers' comments (Reviewer: #2)

Comments to the author:

Very nice work! I am happy with the revision; thus, I recommend publication to Nature Communications.

Response:

We sincerely thank you for your high evaluation, professional guidance and positive recommendation to our manuscript for publication.

Responses to reviewers' comments (Reviewer: #3)

Comments to the author:

The revised paper based on the reviewer's comments is excellent, supported by very substantial quantitative data and experimental results. In particular, it clearly demonstrates the usefulness and novelty of the newly developed robotic hand. The revised paper is worthy of publication because it provides important guidance to many researchers and is also useful to industry.

Response:

Thank you very much for reviewing our revised manuscript again. In particular, you carefully evaluated the modifications and improvements that we made to the manuscript and its supplementary information. Your publication recommendation to our manuscript is much appreciated.